# Resurrecting ancestral antibiotics: unveiling the origins of modern lipid II targeting glycopeptides

Mathias H. Hansen [1,2,3,8], Martina Adamek [4,5,6,8], Dumitrita Iftime[4,8], Daniel Petras[4], Frauke Schuseil [4,5,6], Stephanie Grond [7], Evi Stegmann [4,6] ✉, Max J. Cryle [1,2,3] ✉ & Nadine Ziemert [4,5,6] ✉

Antibiotics are central to modern medicine, and yet they are mainly the products of intra and inter-kingdom evolutionary warfare. To understand how nature evolves antibiotics around a common mechanism of action, we investigated the origins of an extremely valuable class of compounds, lipid II targeting glycopeptide antibiotics (GPAs, exemplified by teicoplanin and vancomycin), which are used as last resort for the treatment of antibiotic resistant bacterial infections. Using a molecule-centred approach and computational techniques, we first predicted the nonribosomal peptide synthetase assembly line of paleomycin, the ancestral parent of lipid II targeting GPAs. Subsequently, we employed synthetic biology techniques to produce the predicted peptide and validated its antibiotic activity. We revealed the structure of paleomycin, which enabled us to address how nature morphs a peptide antibiotic scaffold through evolution. In doing so, we obtained temporal snapshots of key selection domains in nonribosomal peptide synthesis during the biosynthetic journey from ancestral, teicoplanin-like GPAs to modern GPAs such as vancomycin. Our study demonstrates the synergy of computational techniques and synthetic biology approaches enabling us to journey back in time, trace the temporal evolution of antibiotics, and revive these ancestral molecules. It also reveals the optimisation strategies nature has applied to evolve modern GPAs, laying the foundation for future efforts to engineer this important class of antimicrobial agents.

Natural products form one of the most important sources of medicinal compounds, with modern medicine reliant on antibiotics that often originate from biosynthesis in various microorganisms[1]. Indeed, thousands of compounds have been isolated from natural sources—with many more predicted—that display enormous structural diversity. These compounds are mainly produced as so called secondary or specialised metabolites by organisms and represent important adaptive characteristics that have been subjected to natural selection

[1]Department of Biochemistry and Molecular Biology, The Monash Biomedicine Discovery Institute, Monash University, Clayton, VIC 3800, Australia. [2]EMBL Australia, Monash University, Clayton, VIC 3800, Australia. [3]ARC Centre of Excellence for Innovations in Peptide and Protein Science, Monash University, Clayton, VIC 3800, Australia. [4]Interfaculty Institute of Microbiology and Infection Medicine Tübingen, Cluster of Excellence 'Controlling Microbes to Fight Infections', University of Tübingen, Tübingen, Germany. [5]German Centre for Infection Research (DZIF), Partner Site Tübingen, Tübingen, Germany. [6]Institute for Bioinformatics and Medical Informatics (IBMI), University of Tübingen, Tübingen, Germany. [7]Institute of Organic Chemistry, University of Tübingen, Tübingen, Germany. [8]These authors contributed equally: Mathias H. Hansen, Martina Adamek, Dumitrita Iftime. ✉e-mail: evi.stegmann@biotech.uni-tuebingen.de; max.cryle@monash.edu; nadine.ziemert@uni-tuebingen.de

during evolution[2,3]. Given the importance of biosynthetic processes to access complex natural products at scale, understanding how such pathways evolve is crucial information if we are to successfully reengineer such assemblies to allow the formation of new, designer compounds with improved properties.

The biosynthesis of natural products is typically encoded by biosynthetic gene clusters (BGCs), which usually include genes for precursor and core biosynthesis, post-core biosynthesis, regulation, resistance, and transport. Genome analysis has shown that BGCs—and hence natural products—evolve through a range of processes including the recombination of specific subclusters, gene conversion, gene duplication and horizontal gene transfer[4]. However, although some evolutionary models have been proposed[2,5], little is understood about the molecular mechanisms of how natural products, arguably the largest and most economically important source of chemical diversity on the planet, have evolved. Recent exciting work to address this has made use of ancient DNA to investigate natural products from the Pleistocene era[6], although longer evolutionary timescales are doubtless challenging for such an approach.

In this work, we have sought to understand the evolutionary history of lipid II targeting glycopeptide antibiotics (GPAs), a vital class of nonribosomal peptides used in the clinic for the treatment of resistant bacterial infections and exemplified by vancomycin (Van) and teicoplanin (Tei) (Fig. 1)[7,8]. Various types of GPAs are known[9],

which extend beyond the lipid II targeting GPAs under investigation here to type V GPAs such as corbomycin[10] and kistamicin[11] that possess altered structures and modes of action (Fig. 1). All GPAs contain a multicyclic peptide core structure that is assembled through the combined activity of a nonribosomal peptide synthetase (NRPS) machinery[12] and cytochrome P450 monooxygenases, which catalyse a cascade of oxidative crosslinking reactions[13]. The peptide core of GPAs is largely composed of aromatic amino acids including nonproteinogenic amino acids such as 4-hydroxyphenylglycine (Hpg), 3,5-dihydroxyphenylglycine (Dpg), and β-hydroxytyrosine (Bht). Curiously, one key difference across the biosynthetic pathways for lipid II targeting GPAs is the formation of Bht, which is either obtained by tyrosine (Tyr) oxidation on the NRPS as in the teicoplanin (Tei) pathway or generated offline and directly incorporated as in the vancomycin (Van) pathway. Beyond variations in Bht formation and the core peptide, diversity within the GPA family is expanded yet further through modifications to the post-peptide assembly process[7,8].

With the exception of type V GPAs, GPAs function by interrupting bacterial cell wall biosynthesis through the sequestration of the peptidoglycan precursor lipid II[7]. Whilst lipid II targeting GPAs—such as Tei and Van—share a conserved mechanism of action, they differ in the structures of their peptide cores and the BGCs encoding these antibiotics. Earlier phylogenetic reconciliation indicated that the origins of

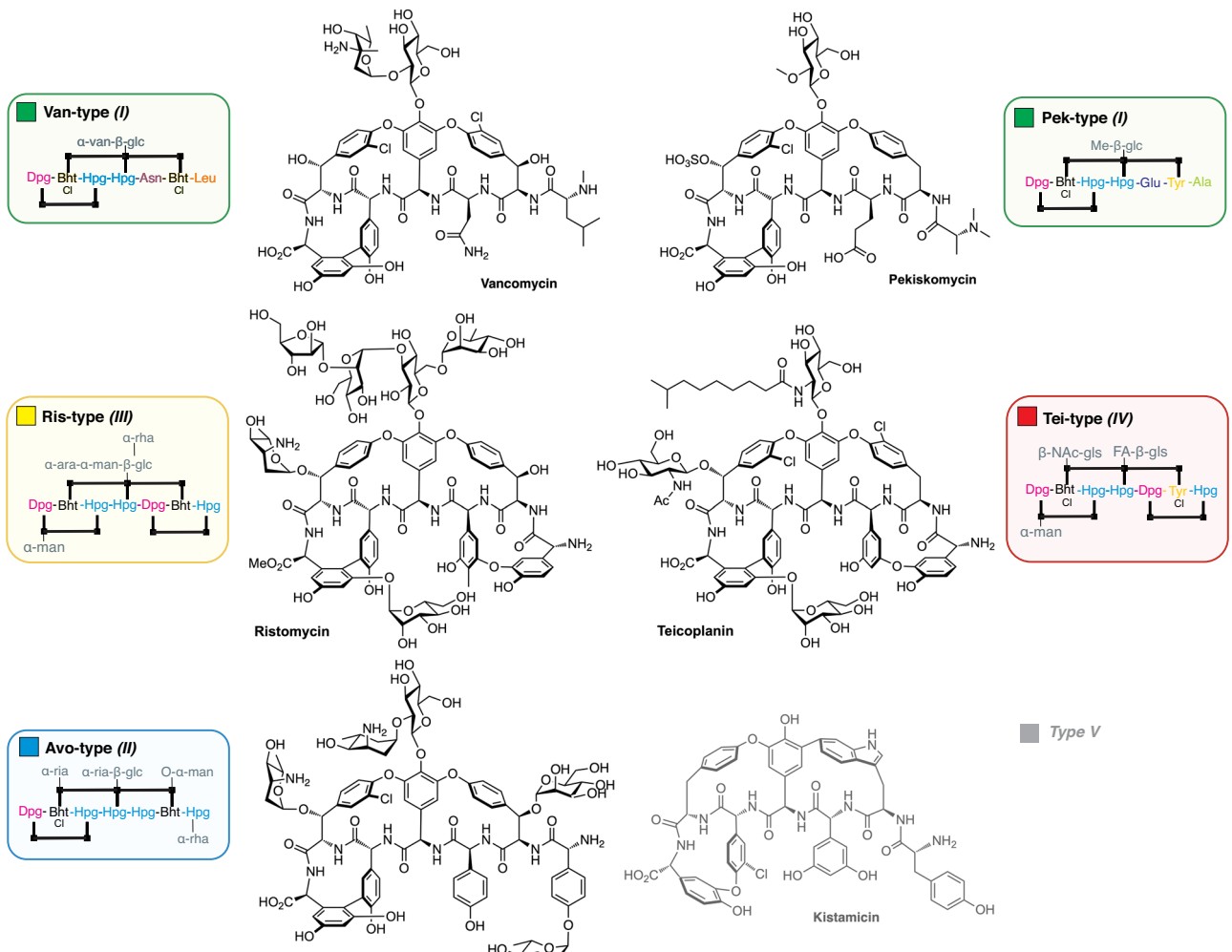

**Fig. 1 | Structures of different GPA types.** Structures of GPAs from all representative types (Van, Pek, Avo, Ris, Tei) shown together with the alternative type I-V nomenclature. Compound abbreviation type GPA naming is used in this manuscript except for the type V GPA outgroup. Sugar abbreviations: D-arabinose (ara), D-glucosamine (gls), D-glucose (glc), 2-O-methyl-D-glucose (Me-glc), D-mannose (man), L-rhamnose (rha), L-ristosamine (ria), L-vancosamine (van). FA (fattyacyl), Ac (acetyl).

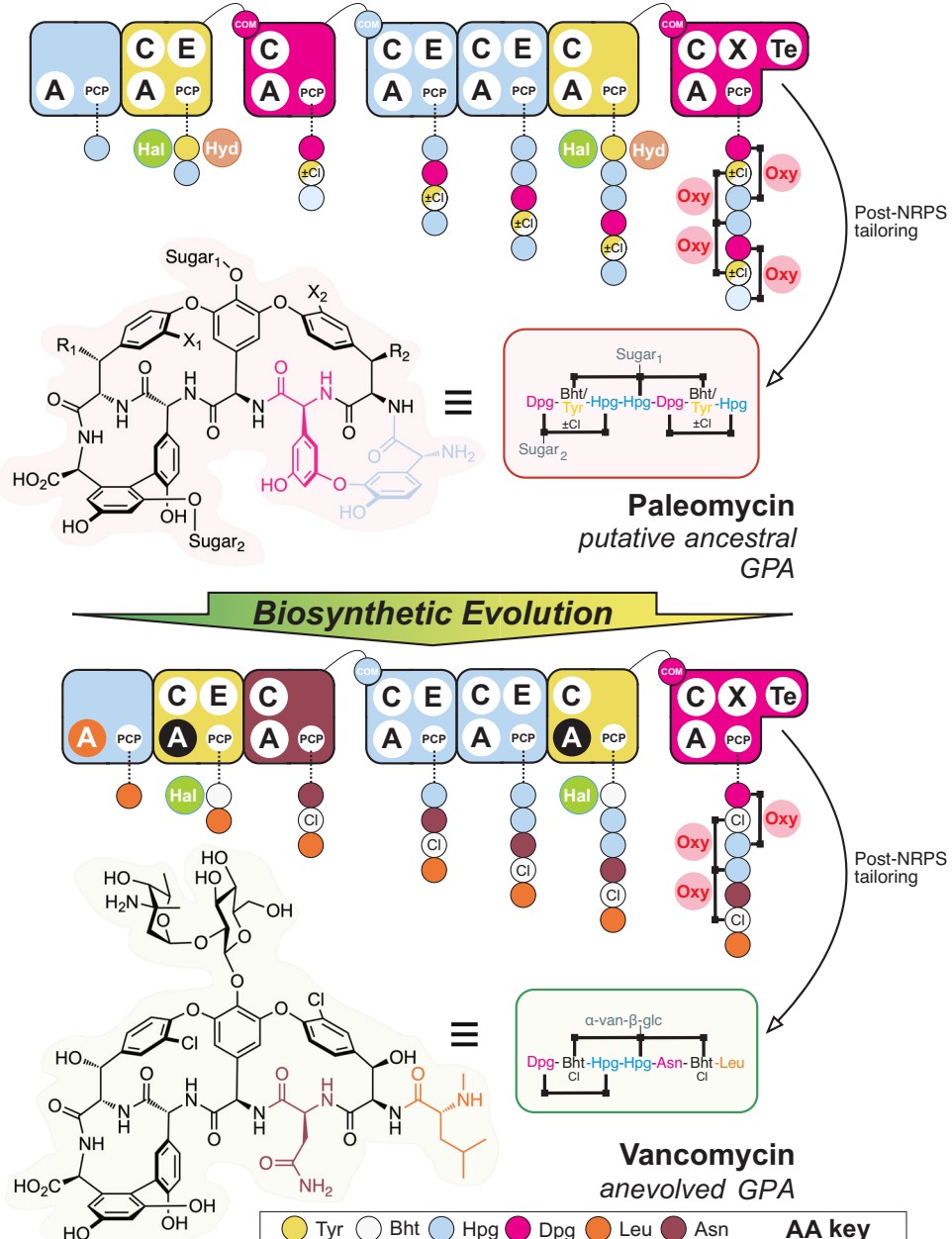

**Fig. 2 | The evolution of GPA biosynthesis from the predicted ancestral GPA paleomycin to modern GPAs such as vancomycin.** Possible modifications to paleomycin inferred from ancestral reconstruction of the BGC include chlorination ($X_1$, $X_2$), hydroxylation ($R_1$, $R_2$) and glycosylation (Sugar$_1$: glucose, Sugar$_2$: mannose, right panel). NRPS assembly lines are shown for each compound, with modules (collections of domains able to install one amino acid into the growing peptide, depicted as rounded rectangles) coloured coded by the amino acid selected (see key). Domain/enzyme description: A adenylation (orange indicates evolution to

Leu selection from Hpg, black indicates evolution to Bht selection from Tyr), C condensation, E epimerisation, TE thioesterase, PCP peptidyl carrier protein, X cytochrome P450 recruitment, COM intermodule communication, Hal halogenase (chlorination, green), Hyd non-heme iron monooxygenase (hydroxylation, brown), Oxy cytochrome P450 (crosslinking, pink). Possible chlorination ($X_1$, $X_2$) during assembly indicated by ±Cl; possible hydroxylation ($R_1$, $R_2$) during assembly indicated by half yellow/white amino acid colouring. Sugar abbreviations: D-glucose (glc), L-vancosamine (van).

glycopeptide biosynthesis can be traced back to a timeframe of around 150–400 million years ago[14].

Our results show that modern lipid II targeting GPAs have evolved from an ancestor—here termed paleomycin—whose predicted core resembles the more complex structure of Tei, suggesting Van-type GPAs are more recent examples of GPA evolution (Fig. 2). We have reconstituted the predicted ancestral NRPS assembly line encoding the paleomycin core peptide, demonstrated production of an antibiotic bearing the core structure of paleomycin from this NRPS, and identified the roles of assembly line recombination and domain mutation in

the generation of modern GPAs. Finally, we have obtained structural snapshots of key selection domains during the evolution of modern lipid II GPAs, providing crucial insights into the general evolution of NRPS-produced peptides.

## Results

### The ancestral lipid II targeting GPA paleomycin is predicted to be a complex peptide structurally related to teicoplanin
To understand the evolution and diversification of genes involved in the biosynthesis of Van/Tei-GPAs displaying lipid II targeting (tricyclic

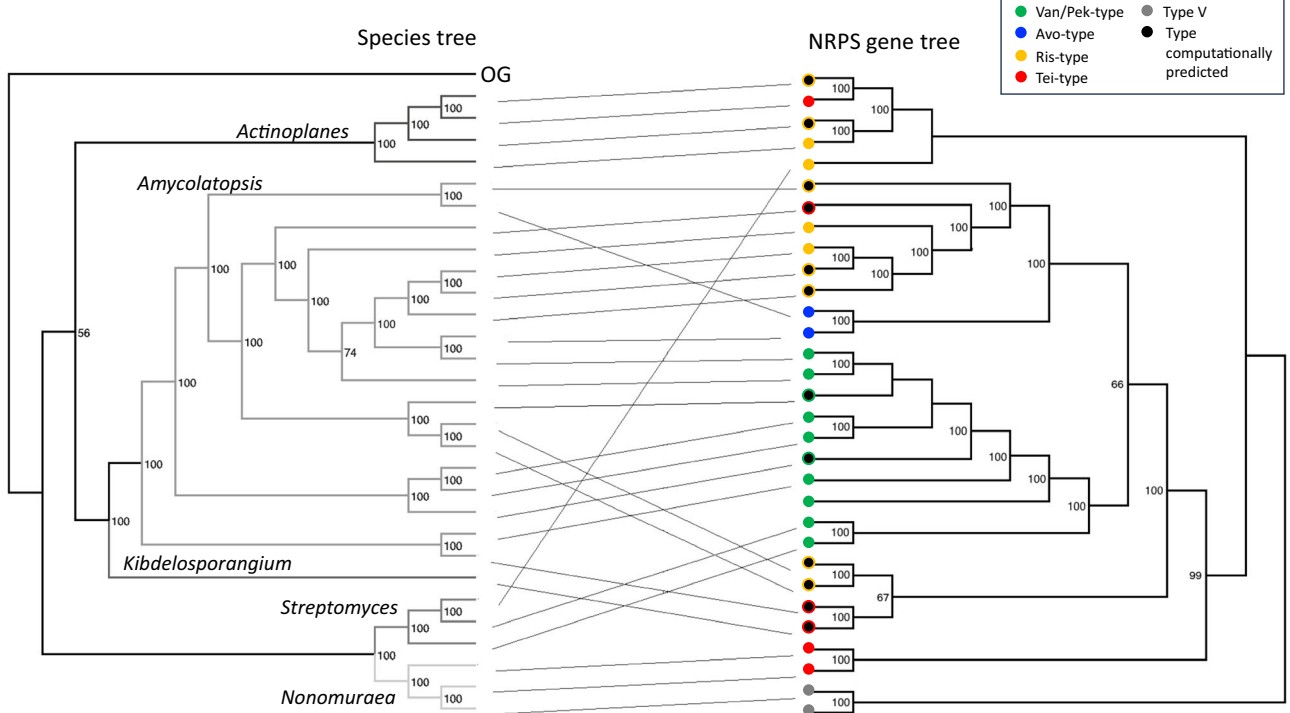

**Fig. 3 | Tanglegram of a species tree of GPA producers versus the gene tree of the respective GPA biosynthesis (NRPS-encoding) genes.** The species tree is a multilocus sequence tree based on concatenated housekeeping genes; *Tistlia consotensis* USBA 355 is used as the outgroup (OG). The types of GPA encoded are colour coded in the GPA gene tree (corresponding to the colours shown for each representative GPA type in Fig. 1). Grey circles represent type V GPAs used as outgroup in the NRPS tree, black circles represent computationally predicted GPAs where the product has not yet been characterised.

(Van/Pek/Avo-type) GPAs; tetracyclic (Ris/Tei-type) GPAs), we generated a guide tree based on 29 complete BGCs (Table S1). In doing so, we deliberately excluded the type-V GPAs whose evolutionary origins have been previously explored by Wright and co-workers[14] and which led to the discovery of the mechanism of autolysin inhibition shown by type V GPAs[10]. Whilst this study also investigated the origins of lipid II targeting GPAs, we felt that the use of a species-centric approach did not sufficiently reflect the evolution of BGCs[4]. We adopted a molecule-centric perspective, treating BGCs as distinct and separate entities, which involved constructing a guide tree to serve as an equivalent of a species tree for the BGCs. Our main goal in doing so was to understand the gene gain/loss and the synchronicity of evolutionary events in the history of BGC evolution, with a focus on the genetic content of the BGCs and the simultaneous changes in the molecular structures of the encoded GPAs. To create the guide tree, we utilised aligned and concatenated GPA-NRPS genes from 29 complete GPA BGCs (Supplementary Fig. S2), which we chose because these NRPS genes offer the most accurate reflection of the BGCs' phylogenetic history. This approach allowed us to establish a BGC species tree in similar manner to traditional species trees that rely on multiple conserved vertically inherited concatenated housekeeping genes. In this context, the most conserved congruent core domains within the GPA BGCs were treated as equivalent to housekeeping genes, enabling us to independently map gene gain and loss of tailoring enzymes, regardless of the bacterial species in which these events occurred.

We observed that specific clades were mixed within different genera, suggesting the occurrence of horizontal gene transfer (HGT) during GPA evolution. This was clearly visible when comparing the guide tree of GPAs with a species tree of GPA producers (Fig. 3).

The resulting tanglegram provided evidence for multiple events of HGT into *Amycolatopsis* and *Streptomyces*. This process was accompanied by major rearrangements of gene synteny/order and explains the production of different types of GPAs by both genera.

With GPA evolution not following species phylogeny, we inspected the main clades of the tree to determine the structure of the encoded GPA. Our analysis revealed that the major clades correlate with the peptide core of the GPA structure they encode (Fig. 4). Furthermore, the major rearrangements coinciding with HGT events do not necessarily result in the production of a new type of GPA. Most curiously, the distribution patterns of different types of GPA within the tree clearly showed that the more complex tetracyclic GPAs (like Tei) are in fact more like ancestral GPAs, while the Van tricyclic GPAs have undergone structural simplification. Ancestral state reconstruction considering the core biosynthesis genes present in GPA BCGs allowed us to predict the genes present in the BGC of the ancestral lipid II targeting GPA—here termed *paleomycin* – and revealing this putative ancestral GPA to be a tetracyclic peptide containing the same proteinogenic (Tyr) and non-proteinogenic (Bht, Hpg, Dpg) residues found in Tei-type GPAs. Ancestral state reconstruction also suggested that generation of the Bht precursor in paleomycin biosynthesis followed the Tei pathway (hydroxylation of NRPS-bound Tyr by a non-heme iron oxygenase (hydroxylase; Hyd)), and further that halogenation (99.9% likelihood) and glycosylation (99.3% likelihood for glucose at Hpg-4 (position 4 of the peptide); 97.5% likelihood for mannose at Dpg-7) occurred during the biosynthesis of paleomycin. The presence of an N-methyltransferase was less certain (53.8%), and thus this was not included in the predicted structure of paleomycin (Figs. S4–6, S8, S10–11, S17–18; S20–24, S26, S29).

**Reconstitution of paleomycin biosynthesis yields an active GPA**
To validate the structural predictions based on our bioinformatic analyses, we next set out to reconstitute the biosynthesis of the peptide core of paleomycin (Fig. S31). The bioinformatically inferred DNA sequences of the ancestral NRPS genes (27.859 kb, *nrps_anc*; with identities between 76% and 85% to the NRPS genes of ristomycin (Table S4)) were synthesised in their entirety by ATG:synthetics (Merzhausen,

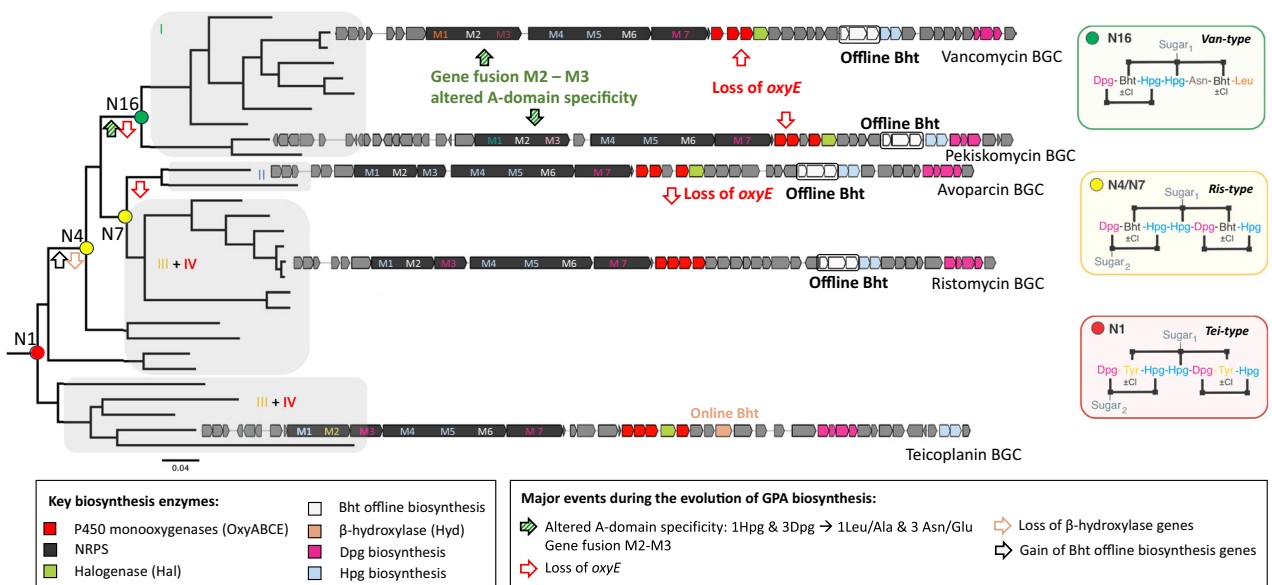

**Fig. 4 | Diversification of the glycopeptide antibiotics (GPAs).** Major diversification events during the evolution of GPA biosynthesis (analysed with ancestral state reconstruction) are indicated on the phylogenetic tree of GPA biosynthetic genes by arrows pinpointing when major structural changes occurred during evolution. The GPA types are indicated (I-IV), with the two clusters of Ris/Tei-type GPAs differing in the mechanism of β-hydroxytyrosine (Bht) incorporation during biosynthesis (online: Tyr hydroxylation by a β-hydroxylase on main NRPS (orange); offline: formation of Bht by hydroxylation of Tyr on a separate minimal NRPS module (white)). Predicted GPA structures indicated for nodes 1, 4/7 and 8. M (module).

Germany) and cloned into the p3SV vector, allowing integration into the surrogate host chromosome. To increase GPA expression levels, we introduced the strong artificial constitutive promoter $Sp44^*$ upstream of the $nrps_{anc}$ genes, resulting in the plasmid pDI1 (Fig. S32). As a chassis for the expression of $nrps_{anc}$, we chose *Amycolatopsis japonicum* MG417-CF17, the producer of the tetracyclic GPA ristomycin (Ris, also known as ristocetin), as this strain possesses genes encoding key biosynthetic enzymes (peptide crosslinking P450 enzymes) as well as enzymes for the biosynthesis of the non-proteinogenic amino acids Hpg and Dpg[15]. In addition, *A. japonicum* MG417-CF17 encodes isoforms of the gatekeepers of the shikimate pathway, 3-deoxy-D-arabino-heptulosonate 7-phosphate (Dahp) synthase and prephenate dehydrogenase (Pdh), which are important to enhance GPA yield and found in most GPA producers. We first engineered *A. japonicum* to selectively remove the ristomycin NRPS genes $rpsA$-$D$ by homologous recombination using the plasmid pGUSA21_RistoKO containing upstream and downstream flanking regions (1.3 and 1.5 kb, respectively) of $rpsA$-$D$ (Figs. S33–S35). To ensure transcription of the remaining genes of the Ris BGC, a second copy of the StrR family regulator gene $bbr^{[15]}$ under the control of the strong constitutive promoter $permE^*$ was integrated into the genome of the *A. japonicum* via the $\Phi C31$ att site.

We next addressed differences in the pathways that lead to the generation of Bht for ristomycin production (Van-type) compared to paleomycin (Tei-type). We replaced the "offline" Van-type Bht forming cassette (including the three genes $oxyD$, $rpsE$ and $bhp$) with that of a Tei-type non-heme iron oxygenase (hydroxylase) from *Nonomurea gerenzanensis* ATCC 39727 (the producer of A40926, the precursor of dalbavancin), which acts "online" directly on the main NRPS (Figs. S36–S38). The absence of a halogenase gene in the Ris cluster was compensated by the introduction of the halogenase gene from *N. gerenzanensis* ATCC 39727[16,17]. Finally, we introduced the plasmid pDI1 via intergeneric conjugation to generate *A. japonicum* DI_$nrps_{anc}$ using this optimised host. Comparative metabolic analysis revealed the presence of a distinct peak in the culture filtrate of *A. japonicum* DI_$nrps_{anc}$ (Figs. S31, S39). The putative GPA was extracted, and tetracyclic Tei-like GPA compounds analysed by liquid chromatography

high resolution tandem mass spectrometry (LC-HR-MS/MS), Fig. 5). Detailed coupled HPLC-MS and MS/MS analysis in combination with molecular networking[18] revealed a set of related ristomycin/paleomycin hybrid GPAs built from a common core paleomycin peptide backbone with distinct structural modifications installed by the Ris host-specific machinery (Figs. 5, S40–41). Inhibition assays confirmed the biological activity of these GPA extracts towards *Bacillus subtilis* (with no activity observed from the negative control, see Fig. S42), demonstrating the biosynthesis of the peptide core of paleomycin by the ancestral NRPS and its successful cyclisation by Oxy enzymes as predicted by bioinformatic analyses.

## The evolution of paleomycin simplifies the GPA scaffold whilst retaining activity
Having revealed the composition of the BGC of paleomycin and the activity of this ancestral GPA, we next investigated the evolutionary pathway towards modern, simplified Van-type GPAs. Ancestral state reconstruction revealed at what stage major changes, including gene gain/loss, gene merger and amino acid exchange, occurred in those BGCs throughout their evolution (Fig. 4). Our analysis showed that significant changes occurred simultaneously in the ancestral node of all Van-type GPAs: two of the seven aromatic amino acids were altered from Hpg-1 and Dpg-3 into proteinogenic aliphatic amino acids Leu-1 and Asn-3 for Van and Ala-1 and Glu-3 for the Van-type GPA pekiskomycin (Pek)[19]. The loss of Hpg-1 and Dpg-3 prevents the typical F-O-G crosslink between these residues, leading to the loss of the OxyE P450-encoding gene from the BGCs of tricyclic Van-type GPAs. The fusion between module M2 and M3 of the NRPS also occurred at this juncture. Another important development in GPA evolution was the change in the biosynthesis of Bht from the Tei-type to the Van-type offline route of Bht supply (Figs. S5–S6). Reconstruction of the evolutionary history of GPA tailoring enzymes showed that these evolved either through duplication or were acquired via HGT, mostly from other Actinobacteria (SI Ancestral State Reconstruction; Figs. S7, S9, S12–16, S19, S25, S27–28). Indeed, gene loss/gain occurred far more often than modifications leading to alterations in the NRPS backbone, which is consistent with the

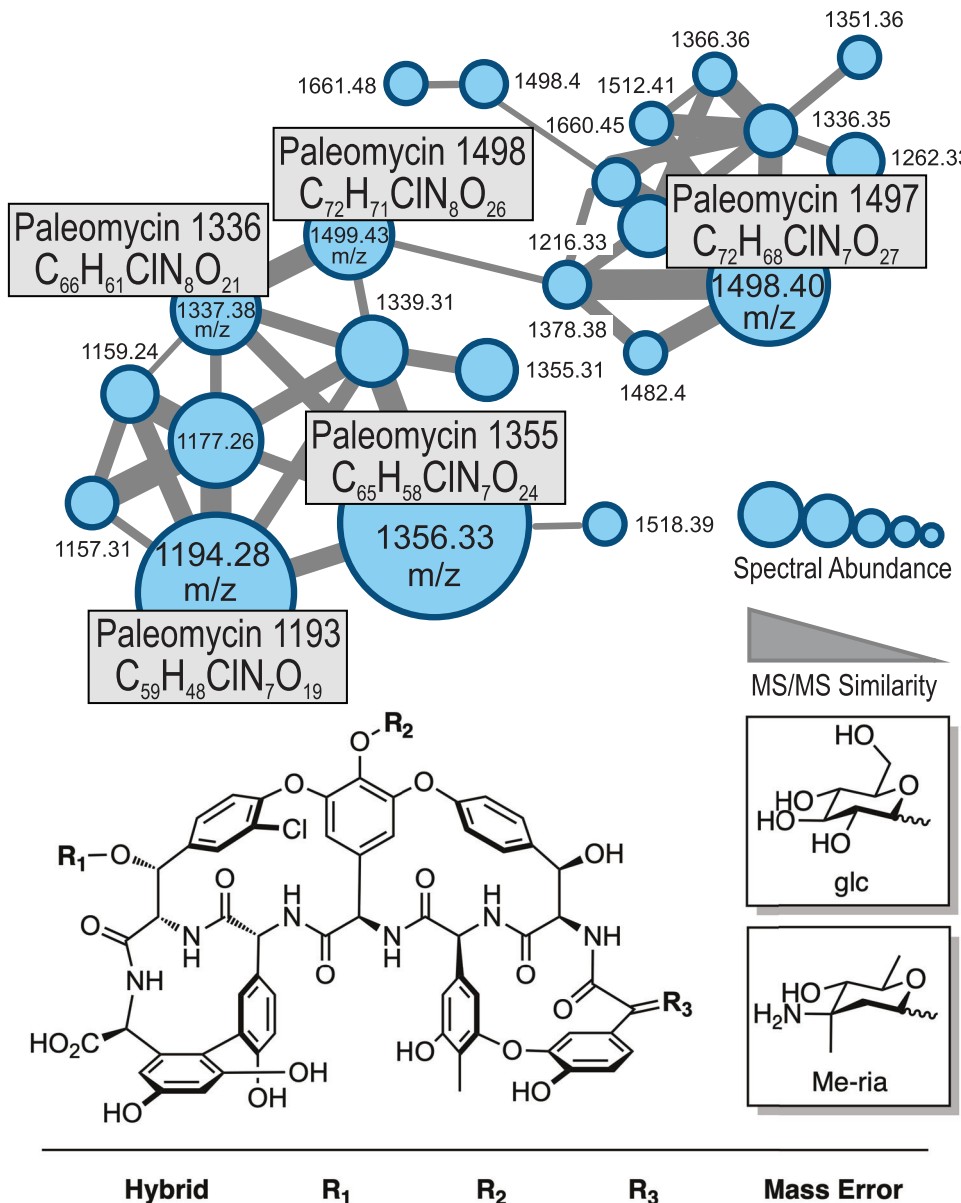

**Fig. 5 | GPA products from the integration of synthetic paleomycin NRPS genes into the modified ristomycin GPA (Ris) producer strain *Amycolatopsis japonicum*.** HR-MS/MS based molecular network analysis[18] of the products of the modified GPA producer confirm the biosynthesis of ristomycin/paleomycin hybrid derivatives comprising the anticipated core peptide, halogenation pattern and glycosylation. Sugar abbreviations: D-glucose (glc), L-ristosamine (ria).

| Hybrid | R$_1$ | R$_2$ | R$_3$ | Mass Error |
|---|---|---|---|---|
| Paleomycin 1193 | H | H | H, NH$_2$ | -3.8 ppm |
| Paleomycin 1336 | glc | H | H, NH$_2$ | -4.0 ppm |
| Paleomycin 1355 | H | Me-ria | H, NH$_2$ | -4.0 ppm |
| Paleomycin 1497 | glc | Me-ria | O | 1.5 ppm |
| Paleomycin 1498 | glc | Me-ria | H, NH$_2$ | -4.2 ppm |

retention of lipid II targeting activity primarily provided by the tricyclic GPA peptide backbone.

Considering the possible implications for engineering approaches, we delved deeper into the events that led to the alteration in the core GPA peptide sequence towards aliphatic amino acids. NRPS assembly lines are highly complex and dynamic[20–23], making rational engineering challenging[24,25]. To understand how the adenylation (A)-domains[26]—key amino acid selection and activation domains that utilise an amino acid selection pocket that can be bioinformatically predicted[27]—found in these NRPS modules changed their specificity towards different amino acids, we performed a phylogenetic analysis of the different GPA modules (Fig. S43), revealing that both the amino acid selection A-domain and peptide bond forming condensation (C)-domain within M3 of tricyclic Van-type GPAs do not clade with other M3 domains[14]. This suggests a different evolutionary origin for these domains—the recombination of M3—likely concomitant with the

fusion of M2 with M3 to overcome the loss of protein/protein interactions resulting from the recombination event. The change in amino acid specificity in M1, however, was predicted to occur by point mutation rather than recombination. To explore these results, we performed sliding window analyses[28] to assess recombination among the GPA NRPS genes (Fig. S30), which clearly showed recombination of C-/A-domains in M3 but not M1 for Van-type GPAs. Further evolution of position 3 of Van-type GPAs is likely (65% probability predicted by ASR) to have occurred from Van to Pek scaffolds, possibly via a second recombination event encompassing only the M3 A-domain. Thus, two different biosynthetic engineering mechanisms have occurred during GPA evolution towards Van-type GPA scaffolds, of which the point mutation strategy for altering A-domain function remained frustratingly opaque (Fig. 44).

## Evolving A-domains as a molecular pathway towards vancomycin production

To explore the process of natural mutation of the A-domain seen in GPA evolution, we assembled an enlarged phylogenetic tree based on the concatenation of the 7 A-domains in each of the 51 NRPSs known or predicted to produce a GPA scaffold (Figs. 6A, B, S45). This larger tree was used to more accurately infer sequence changes that happened at more modern nodes, further removed from paleomycin and closer to extant sequences. The tree topology shows that Van/Pek NRPSs are monophyletic and derived from a single common ancestor. From this

tree, we then selected four key ancestral nodes (ANC1, ANC2, ANC3leu and ANC4) (Figs. 6A, S45) that we hypothesised would unveil the mechanism by which the substrate specificity of the A-domain in the first NRPS module (A1) evolved in tricyclic Van-type GPAs. Ancestral A1-domain codon sequences were inferred from this phylogeny, with the average posterior probability of the reconstructed codon sequences in the four ancestral A-domains ranging from 0.88 to 0.91 (Fig. 46).

Next, we characterised the molecular pathway of GPA evolution by expressing, isolating, and characterising these four ancestral A-domains in terms of their activity towards the anticipated substrates Hpg, Ala and Leu (both D- and L- forms due to the inclusion of a D-configured residue at position 1 despite the lack of E-domain in these modules)[29]. All ancestors were co-expressed with the MbtH-like protein Tcp13, as the presence of a comparable gene in all modern GPA clusters analysed to date implies that such proteins were also necessary and present in the ancestral gene clusters. The expressed domains were all active, displaying $k_{cat}$ values between 0.08–0.89 min$^{-1}$[30], which are comparable to the Tei A1 domain (A1tei; Fig. 6C). ANC1 exhibited an activation rate of 0.75 min$^{-1}$ for L-Hpg, with a fourfold preference for the L-form. ANC2 and ANC3 were both selective for Leu, with ANC3 showing 1.7-/3.8 fold higher $k_{cat}$ value for L-Leu/D-Leu. Activity of ANC2int is consistent with this being an intermediate pocket, showing reduced activity that is improved through natural selection, and supports Van-type GPAs as being older than Pek-type GPAs. ANC4 was highly specific for D-Ala, with an activation rate of 0.89 min$^{-1}$, >15 fold

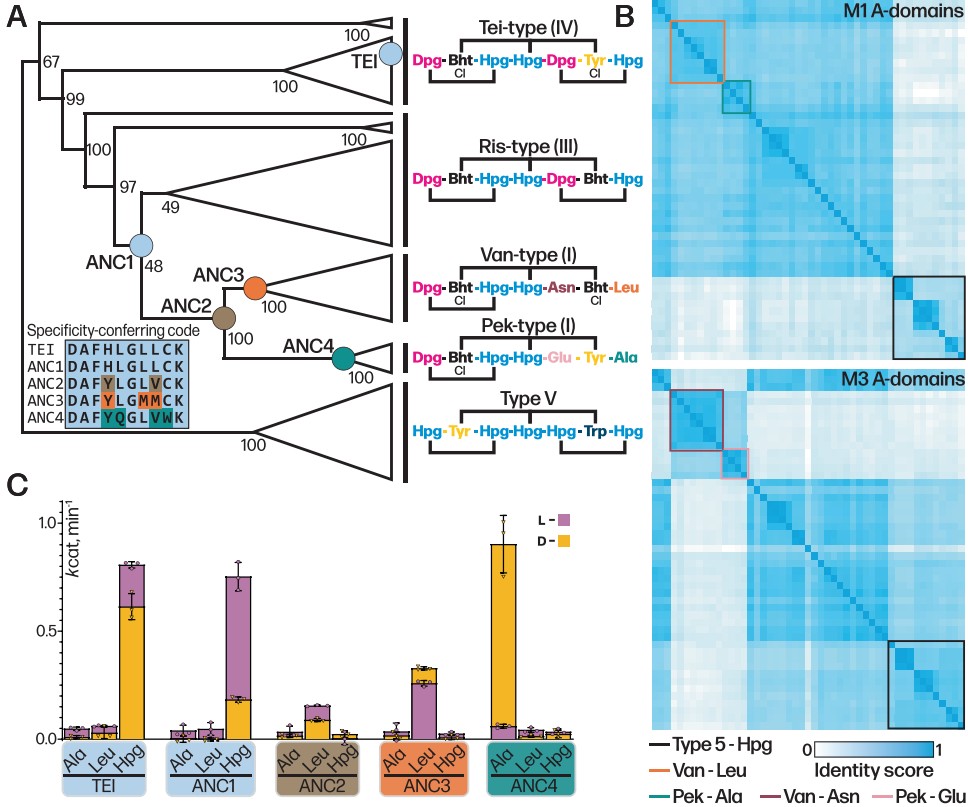

**Fig. 6 | Evolution of M1 A-domain in GPAs has proceeded via point mutation.** **A** Condensed molecular phylogeny of GPA producing NRPSs based on concatenated MSAs of protein and codon sequences of A-domains (note: A-domain from module 6 of the teicoplanin NRPS is coloured as Tyr as this is the AA accepted by this A-domain). Nodes are labelled with bootstrap probability (500 reconstructions). The resurrected ancestral nodes marking the emergence of the Van/Pek scaffold are shown as coloured circles. A-domain specificity-conferring code shows the difference in amino acids aligned to the substrate binding pocket and residue changes are colour-coded to ancestral nodes (ANC1-4). Each clade is annotated with the amino acid substrate of A-domain 1–7 in the assembly line. The

fully annotated phylogeny is presented in Fig. S45. **B** Similarity matrix of A-domains from module (M)1 (top) and M3 (bottom) used in construction of molecular phylogeny. A-domains from Van/Pek scaffold are boxed in orange and cyan for Leu and Ala selecting domains, respectively. A-domains from Van/Pek scaffold are boxed in dark red and pink for Asn and Glu selecting domains, respectively. Type V Hpg selecting domains are boxed in black. **C** A-domain activity of ancestral A-domains ANC1-4 and extant A1tei determined by NADH coupled PPi assay ($k_{cat}$, min$^{-1}$). Data are presented as mean values +/− SEM, $n$ = 3. D- and L-form of amino acid substrate tested is shown as yellow and pink bars, respectively, with the lower activity measurement shown in the foreground of the stacked bars.

higher than activity towards L-Ala. Whilst unexpected, this preference is consistent with D-Ala being present due to cell wall biosynthesis and remodelling of the cell wall that occurs upon GPA biosynthesis[31,32].

To visualise how the ancestral GPA Hpg pocket evolved to select alternate substrates, we turned to structural methods. Since ANC1-3 domains display polydisperse elution profiles in size exclusion chromatography (Fig. 47), we instead turned to the characterisation of an $A_{core}$ construct of the Hpg accepting $A_1$ domain from Tei biosynthesis ($A1_{tei}$), which possesses a homologous pocket to ANC1. $A1_{tei}$ was crystallised together with the MbtH-like protein Tcp13 (PDB ID 8GJ4) to 2.70 Å resolution and also with the substrate L-Hpg to 1.64 Å resolution (PDB ID 8GIC; Fig. 7A, Table S6), revealing a structure highly similar to related enzymes from the adenylate forming enzyme family (e.g., comparison to PheA (1AMU)[33], RMSD of 1.66 Å). The MbtH-like domain Tcp13 also exhibits the anticipated fold, containing a 3-stranded antiparallel β-sheet and an α-helix adjoining the centre of the sheet[34,35].

A365 of $A1_{tei}$ is sandwiched between two (W25 and W35) of the three tryptophan residues found in Tcp13. These tryptophan residues, strictly conserved in MbtH-like proteins, are crucial for complex formation[34,35], and are located at the end of the second β-strand (W25), in the subsequent loop (W35), and in the C-terminal region behind the first α-helix (W54). The Hpg substrate is coordinated by three H-bonds to the α-amino group, one to the side group of D196 and two to the backbone carbonyl groups of L295 and G289. The aromatic ring of L-Hpg is stabilised by hydrophobic interactions with the sidechain of L295 and the main chain of G264, whilst the phenol moiety of L-Hpg is hydrogen bonded to H237 and to the amide nitrogen of G263 (Fig. 7A). The orientation of the H237 imidazole is maintained through three key interactions, two being hydrophobic interactions between the imidazole C5 and L261/L287. A water mediated interaction between E201 and the imidazole Nπ further contributes to imidazole positioning. This Glu residue is widely conserved among NRPS A-domains and serves to

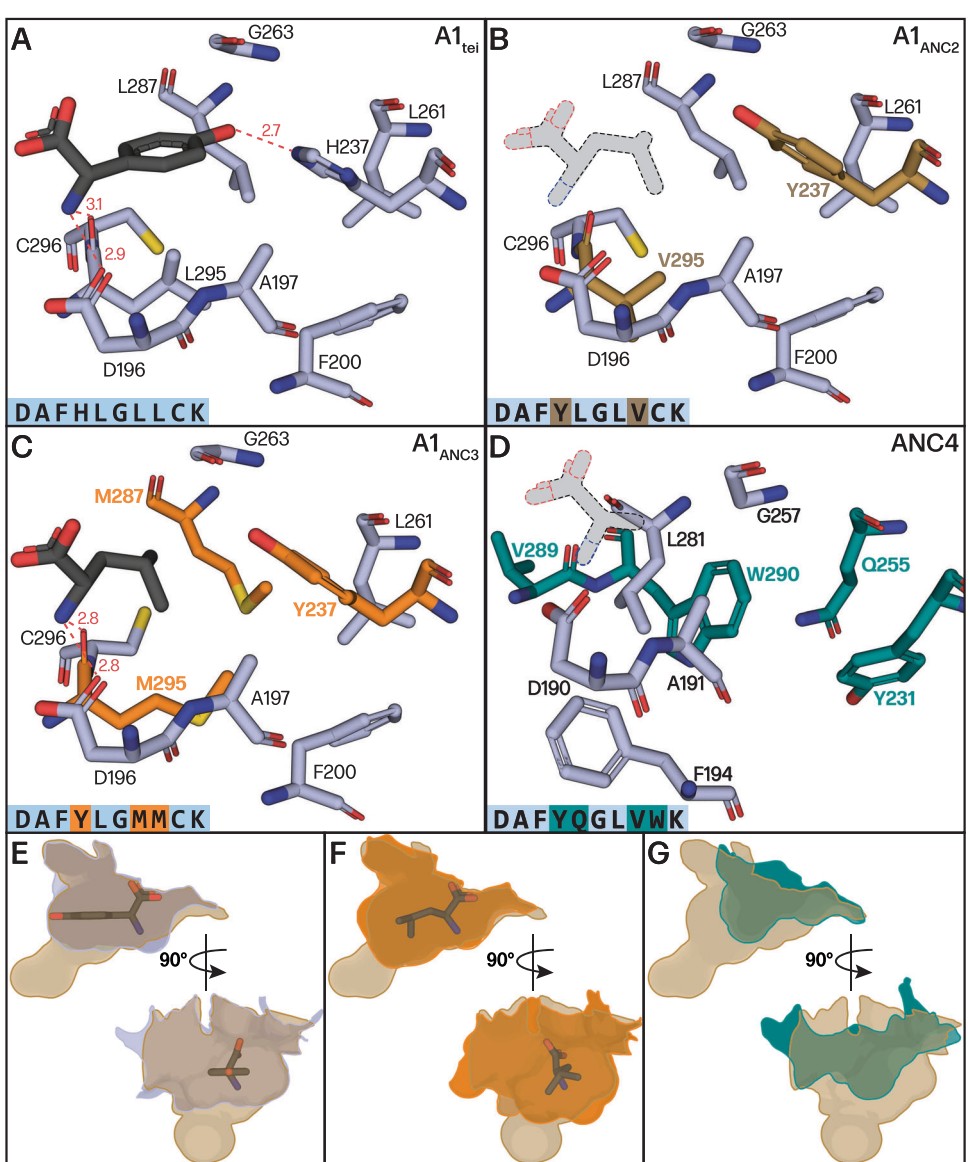

**Fig. 7 | Structures of GPA M1 A-domains during GPA evolution from paleomycin to Van/Pek. A–D** Comparison of substrate binding pockets shown in light blue sticks with residues that have changed between ancestors shown in colour marking ancestral node in phylogeny of Fig. 6 (above) Bound Hpg substrate shown in black for $A1_{tei}$ (which has the same specificity pocket as ANC1) and $A1_{ANC3}$ (Leu). No substrate bound structure was obtained for $A1_{ANC2}$ and ANC4, with substrates shown in stippled lines and greyed out to show an approximate position of bound substrate. **E–G** Superposed and contoured pocket surfaces of $A1_{tei}$, $A1_{ANC3}$ and ANC4 compared with $A1_{ANC2}$, the intermediate pocket, in two views to show how the changes altered the pocket size and shape ($A1_{tei}$ – slate blue, $A1_{ANC2}$ – brown, $A1_{ANC3}$ – orange, ANC4 – turquoise).

maintain the desired orientation of the loop immediately prior to the key α-amino coordinating acidic residue (Fig. 7A, D196 in A1$_{tei}$).

To understand the effects of the evolutionary mutations seen in the binding pocket of the ANC A-domains we then mutated the residues of the substrate binding pocket of A1$_{tei}$ to those seen in A1$_{ANC2}$ and A1$_{ANC3}$. The structures of these mutant domain A$_{core}$ constructs were determined to 2.69 Å and 1.88 Å resolution, respectively (PDB IDs 8GJP and 8GKM; Table S6, Supplementary Fig. S48). Comparison of the A1$_{ANC2}$ pocket with the A1$_{tei}$/ANC1 Hpg pocket revealed that the amino acid substitutions contribute both to prevent Hpg binding and opening the pocket to accept proteinogenic amino acids instead of Hpg (Fig. 7B). The substitution H237Y is key in this process, as it removes H-bonding to the 4-hydroxyl group of Hpg and narrows the pocket to exclude the planar Hpg residue (Fig. 7B). The L295V mutation then opens a cavity that allows the binding of proteinogenic amino acids with a 109° angle at the β carbon. These two key substitutions (H237Y and L295V) enabled binding of new substrates and potentiated the binding pocket for subsequent refinement and partition of substrate binding mode. Comparing the A1$_{ANC3}$ pocket to A1$_{ANC2}$ shows further refinement of the pocket through additional substitutions (L287M and V295M), which serve to improve van der Waals contacts in the binding pocket for Leu (Fig. 7C, Supplementary Fig. S49). Finally, we crystallised the D-Ala accepting ANC4 domain as an A$_{core}$ construct to 3.12 Å resolution (PBD ID 8GLC) because of the instability of this pocket graft into A1$_{tei}$ (Fig. 7D). In this structure, the substrate binding pocket accommodating the small D-alanine is mainly formed by the three residues D196 (numbering for comparison to other structures; D190 in crystal structure), A197 (A191) and L287 (L281). The reduction in pocket size is largely attributed to the C296W (W290) substitution, with the increase in bulk displacing L287 (L281) and shifting the residue upwards by ~3 Å forming the lower surface of the pocket. While Y237 (Y231) is hydrogen bonded to the main chain carbonyl of G263 (G257) in the A1$_{ANC2}$ and A1$_{ANC3}$ pocket, this residue is rotated downwards in ANC4. This rotation appears to be facilitated by a hydrogen bond between the Y237 and the L261Q (Q255) substitution. The new Y237 (Y231) rotamer displaces F200 (F194) and allows for the loop and β-sheet in which G263 (G257) resides to shift downwards, thus further limiting the pocket surface. The ability to interrogate the stepwise changes in the GPA NRPS via mutation of A$_1$ provides important insights into their evolutionary process, showing that the switch from Hpg (ANC1) to Leu (ANC3) did not result in an intermediate pocket (ANC2) capable of activating both residues. Instead, this was a direct selectivity switch that was subsequently refined to provide the modern Van-type GPA A$_1$-domains.

## Discussion

Here, we were able to show that all lipid II targeting GPAs evolved from a common ancestor, predict, and produce the peptide core of this common ancestor and revive an ancestral antibiotic. In doing so, we were able to follow the evolutionary history of GPA diversification on the molecular level and understand both which changes occurred in the past and when these took place. In this regard, understanding the evolution of GPAs is exemplary since they retain a common mechanism of action whilst displaying significant differences in their core peptide structures. Our results show that the ancestor of all lipid-II targeting GPAs, which we term paleomycin, is predicted to possess a complex, tetracyclic heptapeptide core comprised entirely of aromatic residues. These results are in agreement with the results of ref. 14, although different approaches were used to examine the evolution of these molecules. Paleomycin therefore resembles a modern Tei-type GPA, implying that tricyclic, Van-type GPAs containing aliphatic residues have evolved from a more complex precursor. In doing so, evolution of the peptide backbone has been driven by the alteration of the peptide producing NRPS assembly line, with effective concurrent alterations in the first 3 modules (M1-3) of the NRPS, the loss of the 4th

P450 cyclisation enzyme (OxyE) and replacement of Bht generation on the main NRPS (Tei-type) with an offline mechanism for Bht formation (Van-type). Whilst the exact order of these steps can't be precisely defined, biochemical experiments suggest that the presence of OxyE— even in an inactive form—is required to effectively crosslink a peptide containing a Dpg-3 residue[36], suggesting these changes were tightly coupled. Alteration of Bht incorporation could help overcome a lack of A-domain fidelity in M6[37], and prevent the synthesis of GPAs lacking the β-OH group, a crucial attachment site for post-NRPS modification in modern GPAs.

Analysis of the changes in the NRPS from a putative tetracyclic GPA to a tricyclic one indicates the adoption of two reengineering strategies in the evolution of these antibiotics. Our results revealed that both recombination of assembly lines and point mutation of A-domains have occurred during GPA evolution. The replacement of the ancestral GPA M3 by an Asn-encoding module via recombination was observed simultaneously with the fusion event involving M2 and M3. This observation is consistent with experimental evidence that the fusion of M2 and M3 within natural GPA assembly lines results in the subsequent loss of peptide extension[29]. This suggests that GPA evolution overcame the loss of natural inter-module affinity by module fusion. In terms of A1-domain evolution, our results indicate how the alteration of the Hpg binding pocket was driven through three crucial changes in the binding pocket that together eliminated Hpg binding whilst enabling the acceptance of Leu. Optimising mutations then provided the route to the modern M1 A-domains that we see in Van-type GPAs, with Ala-activating domains undergoing further major reengineering to finally activate D-Ala. Structural snapshots of this process demonstrated how major changes in substrate can be accommodated in A-domains, and will no doubt prove key to inform targeted engineering strategies to produce new molecules.

Perhaps the most intriguing question is why GPAs have evolved towards a Van-type scaffold given the limited direct influence that this has had on GPA activity. One possibility is that modern GPAs could have been optimised to generate favourable secondary interactions to further interfere with bacterial cell wall biosynthesis, although Tei-type scaffolds appear to be more robust in this regard[38]. Alternatively, modern GPAs offer a route to improve the titer by tapping into the cellular pool of amino acids more than relying on the shikimate pathway for the supply of building blocks. Indeed, paleomycin requires seven shikimate pathway-derived building blocks whilst vancomycin/pekiskomycin only five[39]. Simplifying the GPA while retaining its specific biological activity might reduce production costs and increase fitness by saving important energetic and molecular resources that can be used for growth and primary metabolism.

In the context of metabolic engineering efforts, this reminds us that from the perspective of the producer any step along the entire biosynthetic apparatus can be optimised and that when engineering assembly lines, the suitable provision of precursors also must be addressed to make these assembly lines effective at producing new peptides[40]. Taken together, our study has revealed paleomycin as the predicted ancestor of all lipid-II targeting GPAs, showing how modern Van-type GPAs have evolved from this Tei-type scaffold through recombination and mutation of the peptide-producing NRPS together with the insertion and deletion of genes encoding precursor generating and peptide modifying enzymes. Our study demonstrates the power of ancestral state reconstruction combined with biochemical and structural approaches to delve into the evolutionary past of antibiotic biosynthesis and understand how changes at the DNA level translate into changes at the enzymatic level to increase peptide diversity. In this way, the evolutionary past of GPAs provides vital insights into the evolutionary mechanisms of secondary metabolism, gives rise to new "ancestral" molecules, and explains how nature's largest chemical diversity unfolded.

# Methods

## Computational techniques

**Ancestral sequence reconstruction of glycopeptide antibiotic (GPA) DNA.** Sequences from all GPA BGCs available by May 2018 (from full genomes and the chloroeremomycin plasmid sequences, see Supplementary Table S1 for strains and accession numbers) were used for the ancestral sequence reconstruction. GPA BGCs were identified using antiSMASH 4[41]. While the actual sequence reconstruction was performed based on nucleotide sequences, the guide tree for ancestral sequence reconstruction was based on protein sequences.

**Horizontal gene transfer of the GPA BGCs.** To infer the evolutionary history of the GPA BGCs we compared the phylogeny of the GPA producer organisms with the phylogeny of the BGCs. A multilocus sequence tree of the producer organisms was generated by autoMLST[42]. A gene tree using the concatenated protein sequences of the GPA BGCs (see "Construction of the guide tree") was compared to the species tree using the tanglegram algorithm in dendroscope 3[43] Fig. 3. Two of branches in the NRPS gene tree are not connected to the species tree, the chloroeremomycin BGC from *Amycolatopsis orientalis* A82846 and the complestatin BGC from *Streptomyces lavendulae*, since the no full genomes were available for these strains. The full trees and tanglegram are available as supplementary data.

**Construction of the guide tree.** For the guide tree protein sequences from NRPS genes containing modules 1–7 were used. For each module, a robust alignment was generated with MAFFT-E-INSi[44] implemented in geneious 9.1.6 (https://www.geneious.com) using the default settings (blosum62 as scoring matrix for the amino acid sequence and a gap open penalty of 1.53). Trimming was performed with GBlocks using the web interface provided by phylogeny.fr at default settings[45,46]. Trimmed alignments for each module were then concatenated. Tree building was performed with RaxML implemented in geneious 9.1.6 using the blosum62 matrix and GAMMA model of rate heterogeneity[47]. A midpoint rooted tree and an outgroup rooted tree, using the NRPS genes from *Nonomuraea sp.* ATCC 55076 (kistamicin) and *Streptomyces lavendulae* (complestatin) as outgroup, were built and compared (Supplementary Fig. S1). Since the midpoint rooted tree only marginally differed from the outgroup rooted tree, and this did not affect tree topology at the node used for ancestral sequence reconstruction, the midpoint rooted tree was used as guide tree for sequence reconstruction and ancestral state analysis. The paleomycin GPA sequence was reconstructed at node N1 (Supplementary Fig. S2).

**Sequence reconstruction.** For the ancestral sequence reconstruction nucleotide sequences form the NRPS genes were used in a codon alignment. Thereby each gene was aligned separately. In case of the van/pek (type I) GPAs modules 1–3 are merged on a single gene. Here modules 1 and 2 were aligned with the first NRPS gene and module 3 was aligned with the second NRPS gene. Furthermore, the sequences of the MbtH-like genes were aligned and used for reconstruction of the MbtH ancestor.

For the sequence alignment three different programs were compared since they were reported to exhibit the best performance for ancestral sequence reconstruction accuracy:[48] (1) MAFFT E-INSi protein alignment used as a basis for codon alignment using a pal2nal.pl script[44,49]. (2) MUSCLE protein alignment used as a basis for codon alignment using a pal2nal.pl script[49,50] (3) Prank codon alignment[51]. All alignment programs were run under default settings.

Ancestral sequences were reconstructed with FastML[52]. Marginal reconstruction was chosen here, since it is considered best for a "sequence-centric" task[48]. JC69 (default) was used as the substitution model for tree inference and sequence reconstruction. As the reconstruction algorithm either considers all or no indels, sequences were manually trimmed according to the following criteria: (i) If insertions were present in only 1–7 sequences, the insertions were removed. (ii) If insertions were present in more sequences, tree phylogeny was considered to make the decision if it was likely that the insertion was part of the ancestral sequence or if it was only present in sublineages that likely evolved later. (iii) If no decision was possible based on the former two criteria, insertions were trimmed in accordance with *A. japonicum* MG417-CF17 as a reference. The best alignment algorithm for sequence reconstruction was chosen after visual inspection of the differences in the three reconstructed sequences. All trees are available at as supplementary data on zenodo (10.5281/zenodo.8410710) [https://zenodo.org/records/8410710].

**Ancestral state reconstruction.** Ancestral state reconstruction was performed for all biosynthesis genes in the GPA clusters that were not conserved in all GPA producer strains. This comprised genes encoding the P450 monooxygenase OxyE, halogenases, glycosyltransferases, acyltransferases, methyltransferases, sulfotransferases, the genes for vancosamine biosynthesis (*evaA-E*), a β-hydroxylase gene, and the three gene cassette to produce β-hydroxytyrosine (*bhp*, *bpsD* and *oxyD*). Regulators, transporters, and resistance genes were not considered. Furthermore, the ancestral states were reconstructed for the NRPS genes considering either the state of three NRPS genes with a GPA backbone of five aromatic and two aliphatic amino acids (type I GPAs) or the state of four genes with a GPA backbone of seven aromatic amino acids (type II-IV GPAs). Ancestral states were reconstructed with Mesquite 3.70, using the maximum likelihood algorithm with a Markov k-state 1 parameter model[53] and the concatenated NRPS tree as guide tree. To identify the closest relatives to the GPA biosynthesis genes, BLAST search was used under default settings, in some cases extending the search to 250 hits[54]. For characterisation of the glycosyltransferases the CAZy (Carbohydrate Active Enzymes) database was used as a reference. [55] Here, GT1 group genes with a characterised function were used to construct a phylogenetic tree. Sequences were aligned using MAFFT e-INS-i. Phylogenetic trees were calculated using IQtree[56]. Thereby substitution models were chosen based on a preceding model test[57] and bootstrapping was performed using UFBoot for ultrafast bootstrapping[58].

**Recombination in GPAs.** To identify recombination in the GPA NRPS genes, the average number of nucleotide differences per site between two sequences ($\pi$) was calculated using the sliding window analysis implemented in DnaSP6[59]. Modules 1–3 of the GPA NRPS genes were pairwise aligned using MAFFT e-INS-i. The window size was chosen to be 100 nt with a step size of 25 nt (Fig. S28), additionally, a window size of 300 nt and the step size 150 nt was analyzed, as previously described by ref. 28 (Supplementary Data).

**Phylogenetics and ancestral sequence reconstruction of A-domains.** A protein multiple sequence alignment (MSA) was generated for the A-domains of each homologous module using MAFFT, with blosum62 as the scoring matrix for the amino acid sequence, a gap open penalty of 1.53, and a gap extension penalty of 0.123. This resulted in seven alignments[44]. The protein alignment was further used to generate a codon alignment through PAL2NAL, a Perl script that converts protein MSAs and corresponding DNA sequences into codon alignments[49]. The resultant MSAs were manually trimmed and concatenated, leading to two MSAs: (I) a protein MSA comprising all A-domains from modules 1–7 for each strain, and (II) the corresponding codon MSA. These two alignments then served as the basis for constructing a phylogenetic tree using IQ-tree version 1.6.3. IQ-tree employs a fast and efficient maximum likelihood tree search algorithm for data set processing[60,61]. For the phylogeny construction, JTT + F + R5 was chosen as the best-fit substitution model, as determined by the in-built model finder in IQ-tree[57]. Bootstrapping was executed with 500 repetitions using UFBoot for ultrafast

bootstrapping[62]. The trees were subsequently visualised with Dendroscope[43].

For ancestral sequence reconstruction, all A-domain sequences from module 1 (Supplementary Fig. S45) underwent realignment using webPRANK—a phylogeny-aware multiple sequence aligner—with the previously generated phylogeny as input[63]. This step was carried out to model gap patterns more accurately, as a phylogeny-aware aligner recognises insertions and deletions as distinct evolutionary events. The phylogenetic tree, together with the webPRANK-aligned protein sequences from module 1, was used for ancestral sequence reconstruction using FastML[52]. The marginal reconstruction employed the "yang" substitution model, which is the default for codon sequences.

### Chemicals and reagents

All strains and plasmids used in this work are listed in Table S2. Oligonucleotides were synthesised either by Integrated DNA Technologies (Leuven, Belgium) or by Sigma Aldrich (Castle Hill, Australia) (Table S3). Sanger sequencing were performed at either the Eurofins Genomics, Ebersberg, Germany or Garvan Molecular Genetics, Darlinghurst, Australia. Q5 Hi-Fidelity DNA polymerase (Thermo Fisher Scientific) or Phusion® High-Fidelity DNA Polymerase (NEB) was used for PCR screening and PCR amplification. PeqGOLD Plasmid Miniprep Kit II and PeqGOLD bacterial DNA Kit (VWR Life Science) were used for plasmid purification and isolation of genomic DNA. In-fusion cloning was performed using the kit from Takara Bio. *Stellar*™ Competent Cells were provided by Takara Bio.

Restriction enzymes are purchased from Thermo Fisher Scientific. Required reagents were achieved from Difco, Sigma-Aldrich, Merck, or Chem Supply.

### Generation and isolation of ristomycin/paleomycin hybrid (Fig. 3I)

**Cultivation of bacterial strains.** *Escherichia coli* Nova blue were used for cloning purposes, and the methylation-deficient strain *E. coli* ET12567 pUB307 was used for intergeneric conjugation.

*Amycolatopsis japonicum*, the ristomycin A producer was used to generate the deletion mutant *A. japonicum DI* (this work)[15].

*E. coli* strains were grown in Luria broth (LB) medium supplemented with 100 μg.mL⁻¹ apramycin or 100 μg.mL⁻¹ hygromycin for selective pressure at 37 °C. Liquid cultures of *A. japonicum/A. japonicum DI/ A. japonicum* p3SV/ *A. japonicum DI_nrps*ₐₙₒ*_bbr_BHH* were cultivated in 50 mL or 100 mL Amycolatopsis production medium (20 g.L⁻¹ glucose, 20 g.L⁻¹ galactose, 10 g.L⁻¹ Bacto Soytone, 2 g.L⁻¹ (NH₄)₂SO₄, 2 g.L⁻¹ CaCO₃ in 1 L water pH 7.4), in 100 mL or 500 mL Erlenmeyer flasks with steel springs at 29 °C and 180 rpm for 3 to 5 days. Liquid/solid media were supplemented with 50 μg.mL⁻¹ apramycin and/or with 25 μg.mL⁻¹ hygromycin to select for strains carrying integrated antibiotic-resistance genes.

**Integration of the *Sp44\** promoter upstream of the ancestral NRPS genes.** The integration of the strong artificial *Sp44\** promoter[64] (Supplementary Fig. S32) was performed by in-fusion cloning using the kit from Takara Bio, according to the manufacturer's protocol. The primers used to amplify the *Sp44\**, SP_OH_Fw/SP_OH_Rv, (Supplementary Table S3), the pDM (p3SV_nrps_anc) was linearised by *EcoRV*.

**Deletion of *rpsA-rspD* in *A. japonicum* to construct *A. japonicum DI*.** To delete the NRPS genes *rpsA-rpsD* the deletion plasmid pGUSA21_Risto_KO was constructed. Therefore, the vector pGUSA21 was used, containing the *gus* (β-glucuronidase) gene as selection marker, the upstream and the downstream flanking regions of *rpsA* and *rpsD*, fragments with sizes of 1.3 and 1.5 kb, respectively. The fragments were amplified by PCR using the genomic DNA of *A. japonicum* as template and the primers Risto_KO_UP fw, Risto_KO_UP rv, Risto_KO_DO fw,

Risto_KO_DO rv containing *NdeI/XbaI* and *SphI/HindIII* restriction sites at the 3′ and 5′ ends (Supplementary Table S3). The PCR amplicons Risto_KO_UP, Risto_KO_DO, were separately introduced into pJET blunt vector yielding pJET_Risto_KO_UP and pJET_Risto_KO_DO. The fragments Risto_KO_UP, Risto_KO_DO were excised from pJET using *NdeI/XbaI* for Risto_KO_UP, and SphI/HindIII for Risto_KO_DO and cloned into pGUSA21, resulting in pGUSA21_Risto_KO. pGUSA21_Risto_KO was transferred into *E. coli* ET12567 pUB307 and finally into *A. japonicum* by intergeneric conjugation. The integration of the plasmid was determined by blue-white screening by plating the transconjugants on MS plates containing 20 mM X-gluc (5-bromo-4-chloro-1H-indol-3-yl-β-D-glucuronic acid). The transconjugants containing pGUSA21_Risto_KO (single crossover mutant) were used for the generation of the in-frame deletion mutant *A. japonicum* DI (S2). To generate an *A. japonicum DI* gene deletion mutant, in which a double crossover event via the second cloned fragment occurred, colonies *A. japonicum_*pGUSA21_Risto_KO were cultivated for two days in 50 ml R5 medium at 29 °C and 180 rpm under apramycin selection. Afterwards, the mycelium was washed and used for inoculation of 50 ml fresh R5 medium without antibiotic selection and cultivated by 37 °C and 180 rpm for 24 h. The cultures were then centrifuged. The mycelium was fragmented by incubation with lysozyme for 15 min and the protoplasts were prepared as described by Matsushima and Baltz[2]. Diluted protoplasts were plated on MS agar plates overlaid with 20 mM X-gluc and white colonies (Supplementary Fig. S33) were picked on new plates. Once double cross-over recombinants were obtained, total DNA was isolated from selected clones and the targeted regions were amplified by PCR (Supplementary Fig. S34).

***Transfer of pIJ_bbr into A. japonicum DI.*** For the overexpression of the pathway-specific StrR-like regulator under the control of the constitutive promoter *ermEp\**[65], the integrative pIJ_bbr vector (Lab Stock)[66] was transferred into *A. japonicum* DI by intergeneric conjugation. The trans conjugants of *A. japonicum* carrying the pIJ_bbr plasmid were selected on hygromycin plates and confirmed by PCR (Supplementary Fig. S35) using the primer pair pIJ fw/pIJ rv.

**Optimisation of *A. japonicum* DI_bbr for heterologous expression of GPA BGCs.** To further optimise *A. japonicum DI_bbr* to produce paleomycin the genes responsible for β-hydroxytyrosine biosynthesis were deleted (Supplementary Fig. 37) and replaced by two genes encoding a β-hydroxylase and a halogenase from *Nonomurea gerenzanensis*, resulting in *A. japonicum DI_bbr_BHH*. To achieve this replacement, we implemented the same strategy as for the deletion of the NRPS genes (Supplementary Fig. S33). To amplify the corresponding fragments, the primers Aj-32060 fw, Aj-32060 rv, BHH-fw, BHH-rv, Aj-32040 fw and Aj-32040 rv were used (Supplementary Table S3, Supplementary Fig. S36). The β-hydroxylase and the halogenase genes were amplified from a previously constructed plasmid in our lab (pBHH).

***Transfer of p3SV and the pDI1 into A. japonicum DI/ A. japonicum DI_bbr_BHH.*** The transfer of the empty plasmid p3SV and pDI1 (harbouring the ancestral *nrps* genes (*nrps*ₐₙₒ) under the control of the SP44\*), respectively, into *A. japonicum DI/A. japonicum DI_bbr_BHH* was carried out using a standard protocol for intergeneric conjugation in actinomycetes. The empty plasmid p3SV and pDI1 (p3SV containing the ancestral NRPS genes (*nrps*ₐₙₒ; the 30 kb synthetic construct) under the control of the constitutive promoter *Sp44\**), were transferred into the methylation-deficient strain *E. coli* ET12567 pUB307 and finally into the chromosomes of *A. japonicum DI/A. japonicum DI_bbr_BHH* via the ΦC31 attB sites by intergenetic conjugation. Overnight cultures of the donor *E. coli* strains and fresh mycelium of the recipient strains were combined in microcentrifuge tube and mixed by pipetting. The mixture was plated on non-selective plates containing MgCl₂ and incubated overnight

at 29 °C. The recombinant mutants *A. japonicum*_p3SV (negative control) and *A. japonicum* DI_nrps_anc_bbr_BHH (with integrated pDI1) were selected on apramycin plates and confirmed by PCR using the primer pair Apra fw/Apra rv for the p3SV (Fig. 38a) and the primer pair PacI GPC_fw/bla vec_rv for pDI1 (Figs. 37b; Supplementary Table S3).

**Production of ristomycin/paleomycin hybrid GPA in *A. japonicum* DI_nrps_anc_bbr_BHH.** The negative control *A. japonicum* DI_p3SV and *A. japonicum* DI_nrps_anc_bbr_BHH were grown for 3 days on petri dishes containing 30 ml of MS-agar. For the cultivation in liquid cultures 1 cm² mycelium was scraped and used to inoculate 50 mL TSB medium as a preculture. 5 mL of 3 days-old preculture were used for inoculation of 100 mL Amycolatopsis production medium. Fermentations were carried out for 5 days at 29 °C in a rotary shaker at 180 rpm. Cultures were centrifuged at 6441 g for 15 min and the culture filtrates were used for HPLC-MS and MS-MS measurements.

**Isolation and purification of paleomycin/ristomycin hybrid GPA.** The cultivation and extraction protocol were scaled up to 1 L Amycolatopsis production medium, which was distributed to 10 Erlenmeyer flasks. Purification was accomplished by adsorption chromatography on XAD16 resin (200 mL). The column was washed with $H_2O$ (800 mL), MeOH (20%), and MeOH (30%). The GPA was eluted with MeOH (800 mL, 100%) and concentrated by evaporation. This step was followed by size-exclusion chromatography on Sephadex LH20. All purification steps were monitored by HPLC-ESI-MS.

**Detection of paleomycin/ristomycin hybrid GPA by HPLC-MS.** The production of paleomycin was detected using an Agilent 1200 HPLC System (Agilent Technologies, Waldbronn, Germany) coupled to an LC/MSD Ultra Trap System XCT 6330, Agilent Technologies, Waldbronn, Germany). Chromatographic separation was performed at a flow rate of 400 µL.min⁻¹ using stationary phase C18 column Nucleosil 100 3 µm (100 × 2 mm ID, fitted with a precolumn 10 × 2 mm, same stationary phase, Dr. Maisch GmbH, Ammerbuch) with the mobile phase composed of formic acid (A = 0.1%), and formic acid in acetonitrile (B = 0.06%). A gradient from 0 to 100% of B in 15 min with a 2-min hold at 100% for solvent B, was used.

**Biological activity assays.** Biological activity of the *A. japonicum* DI_nrps_anc_bbr_BHH culture filtrate was investigated by a disc diffusion assay. The paleomycin/ristomycin hybrid GPA containing solutions and controls (25 µL) were spotted on agar plates containing the indicator strain *B. subtilis* (Supplementary Fig. S42).

**LC-HR-MS/ MS analysis of ristomycin/paleomycin hybrid**
**Mass spectrometry data acquisition.** For UHPLC-MS/MS analysis 2 µL of the samples were injected into vanquish UHPLC system coupled to a Q-Exactive HF quadrupole orbitrap mass spectrometer (running Q Exactive HF Tune 2.12, Thermo Fisher Scientific, Bremen, Germany). For reversed-phase chromatographic separation, a C18 core-shell micro-flow column (Kinetex C18, 50 × 1 mm, 1.8 um particle size, 100 A pore size, Phenomenex, Torrance, USA) was used. The mobile phase consisted of solvent A ($H_2O$ + 0.1% formic acid (FA)) and solvent B (acetonitrile (ACN) + 0.1% FA). The flow rate was set to 150 µL/min (setup A) or 100 µL/min (setup B). A linear gradient from 5–50% B between 0–8 min and 50–99% B between 8 and 10 min, followed by a 3 min washout phase at 99% B and a 5 min re-equilibration phase at 5% B was used.

Data Independent Acquisition (DIA) and Data-dependent acquisition (DDA) of MS/MS spectra was performed in positive mode. Electrospray ionization (ESI) parameters were set to 40 arbitrary units (arb. units) sheath gas flow, auxiliary gas flow was set to 10 arb. units and sweep gas flow was set to 0 AU. The auxiliary gas temperature was set to 400 °C. The spray voltage was set to 3.5 kV and the inlet capillary

was heated to 320 °C. S-lens level was set to 70 V applied. MS scan range was set to 800–2000 m/z with a resolution $R_{m/z\,200}$ of 45,000 or 240,00 with one micro-scan. The maximum ion injection time was set to 100 ms with automatic gain control (AGC) target of 5E5. Either two or five MS/MS spectra per duty cycle were acquired at R $_{m/z\,200}$ 15,000, 120,000, or 240,000 with one micro-scan. The maximum ion injection time for MS/MS scans was set to 100 ms with an AGC target of 5.0E5 ions and a minimum of 5% AGC. The MS/MS precursor isolation window was set to m/z 1. The normalised collision energy was set to 20 or 25% with z = 1 as the default charge state. MS/MS scans were triggered at the apex of chromatographic peaks within 2 to 15 s from their first occurrence. Dynamic precursor exclusion was set to 5 s. Ions with unassigned charge states were excluded from MS/MS acquisition, as well as isotope peaks.

**Mass spectrometry data analysis.** Raw data conversion and peak picking were performed with MSconvert. Centroided data was visualised and manually inspected using the GNPS dashboard (https://dashboard.gnps2.org/)[67]. Classic Molecular Networking was generated with the GNPS platform (gnps.ucsd.edu) using default settings[18] other than precursor and product ion tolerance, which were both set to 0.01 m/z. The Molecular Network was the visualised in Cytoscape[68] and connected MS/MS spectra were manually interpreted. Exact masses and isotope patterns were calculated using EnviPad (www.envipat.eawag.ch) and manually compared using the raw profile data in Qual-Browser (Thermo Fisher, Bremen Germany).

**In vitro characterisation and crystallisation of GPA A-domains**
**Cloning of A-domain constructs from teicoplanin module 1.** A-domains comprise two domains ($A_{core}$ and $A_{sub}$); as the binding of amino acid substrates occurs within the $A_{core}$ domain at the $A_{core}$-$A_{sub}$ interface, two constructs were designed for each A-domain: a full-length A-domain containing both the $A_{core}$ and $A_{sub}$ domains for biochemical characterisation, and an A-domain lacking the flexible 10 kDa C-terminal $A_{sub}$ for crystallography. The sequence for $A1_{Tei}$ was amplified by PCR using Phusion® High-Fidelity DNA Polymerase (NEB) from a synthetic gene encoding the enzyme Tcp9 (Uniprot ID Q70AZ9, Supplementary Table S3) that had been synthesised and codon optimised for expression in *E. coli* by Eurofins Genomics, Ebersberg, Germany. The primers (Supplementary Table S3) were designed to create overhangs compatible for integration the pHis17 vector using In-Fusion cloning (Takara). The In-Fusion reaction and PCRs were performed as indicated by the manufacturer's protocols. The pHis17 vector was linearised using primers pHIS17 fw and pHIS17 rv. The primers A1_tcp9 fw and A1_tcp9 rv or A1_tcp9 fw with A1core_tcp9 rv were used to generate $A1_{Tei}$ (residues 9-492) and $A1_{core-Tei}$ (residues 9-398) amplicons, respectively. The PCR amplicons were separately introduced into pHis17.

**Cloning of ancestral A-domains.** Sequences derived from ASR (Supplementary Table S3) were codon optimised for *Escherichia coli* BL21 (DE3) cells (Novagen) and synthesised in single fragments by integrated DNA technologies (IDT). Fragments were cloned into a modified version of the pOPIN-S vector, comprising an N-terminal hexahistidine-SUMO (Small Ubiquitin-like Modifier) tag and a C-terminal STREP tag, using In-Fusion® HD Cloning Plus (Takara). The primer pairs ANC1 fw + ANC1 rv, ANC2 fw + ANC2 rv, ANC3 fw + ANC3 rv, and ANC4 fw + ANC4 rv were used for amplification of ANC1 to ANC4, respectively. Meanwhile the pOPIN-S vector was linearised using the primer pair pOPIN-S_STREP fw + pOPIN-S_STREP rv. The ANC4_A1_core (residues 1–391) fragment was amplified using the primers ANC4_A1_core fw and ANC4_A1_core rv and integrated into pHis17 using the linear vector backbone generated using primers pHIS17 fw and pHIS17 rv.

**Cloning of pocket graft A_core-constructs.** To introduce the residues mutated in the substrate binding pocket of ANC2 and ANC3 into the $A1_{core\text{-}tei}$ construct, we used In-Fusion cloning. The $A1_{core\text{-}tei}$ vector was linearised using primers $A1_{core\text{-}tei}$_graft fw and $A1_{core\text{-}tei}$_graft rv. Two fragments with compatible overhangs and either the H237Y and L295V mutations for $A1_{core\text{-}ANC2}$ or the H237Y, L287M and L295M mutations for $A1_{core\text{-}ANC3}$ was synthesised as single fragments by IDT and used for the In-Fusion reaction.

**Protein expression and purification of A-domains.** All A-domain constructs were co-transformed with a pCDF-1b construct encoding the MbtH-like protein Tcp13 from *Actinoplanes teichomyceticus* (Uniprot ID: Q70AZ5) into *E. coli* BL21 *(DE3)* cells. Cells were grown overnight at 37 °C with shaking in lysogeny broth (LB) medium (Miller et al. 1992) supplemented with 100 μg/mL ampicillin and streptomycin as a preculture. To start cultivation, the preculture was pelleted and resuspended in fresh LB before inoculation into ten 2 L Erlenmeyer flasks containing 1 L of LB medium per flask (supplemented with 100 μg/mL ampicillin and streptomycin). Cultures were grown at 37 °C until an $OD_{600\ nm}$ of 0.5 was reached. Subsequently, the temperature was decreased to 18 °C and protein expression was induced with isopropyl β-D-1-thiogalactopyranoside (IPTG) at a final concentration of 0.2 mM. Cells were harvested after 20 h of growth by centrifugation at 6000 g for 30 min. The cells were subsequently resuspended in lysis buffer (300 mM NaCl, 20 mM imidazole, 20 mM Tris–HCl, pH 8.0) supplemented with EDTA-free protease inhibitor (*Sigma-FAST™ Protease Inhibitor* Cocktail Tablet) and subsequently lysed by sonication. The total lysate was clarified by centrifugation at 12,000 g for 30 min at 4 °C prior to protein purification. The supernatant was loaded onto a nickel-chelating column (HisTrap Fast Flow crude, 5 mL, GE Healthcare) pre-equilibrated in lysis buffer using an Akta Pure Protein Purification System (running Unicorn 7, Cytiva). The target protein was eluted in fractions with a linear gradient from 20 mM to 300 mM imidazole over 20 column volumes. The fractions collected were analysed by SDS-PAGE and the purest fractions were collected and concentrated using an ultra-centrifugal filter (Amicon Ultra−15) with a molecular weight cut-off (MWCO) of 30 kDa. The concentrated fractions were further purified by size-exclusion chromatography in SEC buffer (20 mM Tris–HCl, pH 8.0 and 300 mM NaCl) using a Sephacryl S-300 hr 16/60 (GE Healthcare) column for $A1_{tei}$ constructs or an SRT-10 SEC-300 (SEPAX) column for ancestral A-domains. The fractions collected were analysed by SDS-PAGE. Fractions containing monomeric protein were chosen based on elution profile and SDS-PAGE gel. Every step of purification was performed either on ice or at 4 °C. Final purified proteins were then concentrated to a minimum concentration of 20 mg/mL using centrifugal filtration (Amicon Ultra−15) with a molecular weight cut-off (MWCO) of 30 kDa before being snap-frozen in liquid nitrogen and stored at −80 °C.

**X-ray crystallography.** To identify crystallisation conditions, initial broad matrix screens were performed at the Monash Molecular Crystallisation Platform (MMCP). Crystals of Tcp9 $A1_{core\text{-}Tei}$ in complex with MbtH-like protein Tcp13 were obtained at a concentration of 10 mg/mL in drop D3 (0.1 M MMT buffer, pH 6.0, with 25% (w/v) PEG 1500) of the PACT premier crystallisation screen (Molecular Dimensions) at 20 °C using the sitting drop vapour diffusion method. Crystallisation drops for all four A-domain complexes contained 1 μL of protein solution mixed with equal amounts of precipitant and were equilibrated against 300 μL of precipitant solution containing the following: (i) for Tcp9 $A1_{core\text{-}tei}$/Tcp13, drop D3 from the PACT premier screen (Molecular Dimensions); (ii) for Tcp9 $A1_{core\text{-}ANC2}$/Tcp13 and Tcp9 $A1_{core\text{-}ANC3}$/Tcp13, drop D12 (0.04 M Potassium phosphate monobasic, 16% (w/v) PEG 8000, and 20% (v/v) Glycerol) from the JCSG+ screen (Molecular Dimensions); and (iii) for $ANC4_{core}$/Tcp13, drop H4 (1.6 M Magnesium sulfate and 0.1 M MES, pH 6.5) from JBScreen Classic HTS II screen (Jena Bioscience).

To obtain substrate bound structures, the crystals were soaked for 2–5 min in the reservoir solution supplemented with 30% sucrose and 4-10 mM L-Hpg, L-Leu or D-Ala before they were mounted on cryoloops and vitrified in liquid $N_2$ prior to X-ray data collection. High-resolution synchrotron diffraction data at 100 K were collected on the MX2[69] beamline at the Australian Synchrotron (Clayton, Victoria, Australia) equipped with an Eiger detector (Dectris).

Data were processed and scaled using the routines XDS, Pointless, and Aimless from the CCP4 suite[70]. During the refinement process, 5% of the reflection data was set aside as R_free for cross-validation and was not used during any stage of the refinement. We applied a high-resolution cut-off criterion of $CC_{1/2} > 0.3$. Data collection statistics are listed in Table S3. The phases for structure determination were obtained by molecular replacement using PHASER from the PHENIX package and the structure of phenylalanine activating domain of gramicidin synthethase 1 from *Brevibacillus brevis* (PDB ID: 1AMU)[33] as a search model for 8GJ4. Using the Phaser solution obtained (two dimers of Tcp9 $A1_{core\text{-}tei}$/Tcp13). We utilised PDB tools to randomise ADPs and conducted Cartesian-simulated annealing with PHENIX to avoid phase model bias and as recommended by the developers[71]. The final refined model was generated using iterations between manual real-space refinement in COOT[72] and automated refinement in PHENIX[71]. Initial stages of refinement primarily involved manual rebuilding, employing basic refinement options such as reciprocal and real space refinement and individual atom isotropic B factors with default NCS restraints. NCS was applied between $A_{core}$ chains A and B, and between the MbtH-like protein chains C and D. X-ray data for geometry weights and atomic displacement factors were automatically determined using the "optimise X-ray/stereochemistry weight" and "optimise X-ray/ADP weight" functions, respectively. Model validation was carried out using COOT[72] and MolProbity[73]. Statistics were generated in PHENIX through the "Generate Table 1 for journal" function. Chain A and D from final model of 8GJ4 was used as search model for 8GIC, 8GJP and 8GKM during molecular replacement followed by the same refinement strategy as for 8GJ4.

**NADH coupled pyrophosphate assay.** PPi release assays[30,74] were performed at 30 °C and the data collected using a V-650 spectrophotometer (running SpectraManager II, Jasco). Each reaction was performed in a total volume of 500 μL. The assay buffer (50 mM Tris-HCl pH 7.4, 10 mM $MgCl_2$ and 0.01 mM EDTA) was supplemented with 1 mM D-fructose-6-phosphate, 0.1 U mL$^{-1}$ fructose-6-phosphate kinase, pyrophosphate- dependent (*Propionibacterium freudenreichii (shermanii)*), 1 U mL$^{-1}$ aldolase, 5 U mL$^{-1}$ triosephosphate isomerase, 5 U mL$^{-1}$ glycerophosphate dehydrogenase and 0.2 mM NADH. To measure the activity of excised A-domains, 10 μM enzyme was added to the assay buffer together with 0.5 mM ATP. Reactions were incubated for 5 min and the reaction was then commenced by the addition of 1 mM substrate. All assays were performed in triplicates. Slopes were fitted using the SpectraManager II software and the fitted data was analysed using GraphPad Prism 8. Velocity was calculated from the slope of the linear phase using the Beer– Lambert law ($v$ = slope (Abs/min)/ ($\varepsilon_{340}$(NADH)*$l$*2).

**Reporting summary**

Further information on research design is available in the Nature Portfolio Reporting Summary linked to this article.

## Data availability

The crystallography data generated in this study have been deposited in the protein data bank (PDB) database under accession codes 8GJ4,

8GIC, 8GJP, 8GKM and 8GLC. The mass spectrometry raw (.raw) and centroided (.mzML) data have been deposited in the MassIVE repository (massive.ucsd.edu) under accession codes MSV000091083 (https://doi.org/10.25345/C5H41JX2M). Supporting data (paleomycin sequence, sequence alignments, phylogenetic trees, bioinformatic scripts, posterior probabilities for the ancestral sequence reconstruction, sliding window analysis) have been deposited at Zenodo under the following accession code 8410710. The sequences of the synthetic genes encoding the ancestral A-domains ANC1-4 and uncropped gel images generated in this study are provided in the Source Data file. Source data are provided with this paper.

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

## Acknowledgements

This research was undertaken in part using the MX2 beamline at the Australian Synchrotron, part of ANSTO, and made use of the Australian Cancer Research Foundation (ACRF) detector[69]. We would like to thank the Monash Macromolecular crystallisation facility, G. Kong for assistance with crystal screening experiments; the MX2 beamline scientists at the Australian Synchrotron for their support during data collection; Thierry Izoré and Hussain Bhukya for support with X-ray crystallography; Mohamed Alanjary for bioinformatics support; Thomas Schaffhauser for providing the vector p3SV; Libera Lo Presti, Wolfgang Wohlleben and colleagues for their valuable feedback on the manuscript. This work was supported by Monash University, EMBL Australia and the Australian Research Council's Discovery Projects funding scheme (project number DP210101752) to MJC. This research was conducted by the Australian Research Council Centre of Excellence for Innovations in Peptide and Protein Science (CE200100012) and funded by the Australian Government (MJC). NZ and MA thank the German Centre for Infection Biology (DZIF TTU 09.716) and the Deutsche Forschungsgemeinschaft (DFG, German Research Foundation) for funding under Germany's Excellence Strategy—EXC 2124—390838134. DI, ES, and DP thank the Deutsche Forschungsgemeinschaft (DFG, German Research Foundation) for funding under Germany's Excellence Strategy—EXC 2124—390838134.

## Author contributions

M.H.H. performed all biochemical and structural biology experiments and analysed the A1 domain phylogeny. M.A. performed the phylogenetic analyses and computational sequence reconstruction of paleomycin, with help from F.S. D.I. performed molecular cloning experiments, cultivated bacterial strains, and isolated the paleomycin/

ristomycin hybrid. S.G. and D.P. performed mass spectrometry analysis and data analysis. E.S. designed and conceived the study. N.Z. designed and conceived the study and performed phylogenetic analysis. M.J.C. designed and conceived the study, assisted with structural and biochemical analysis, and wrote the manuscript. All authors contributed to analysis and revision of the manuscript.

## Competing interests

The authors declare no competing interests.
