## [Peer Review File · Nature Communications]

Resurrecting Ancestral Antibiotics: Unveiling the Origins of Modern Lipid II Targeting GlycopeptidesReviewer #1 (Remarks to the Author):

Hansen et al. have investigated the evolution of glycopeptide antibiotics which are important drugs of last resort. From various related gene clusters, a phylogeny has been built and an ancestral gene cluster for the biosynthesis of the hypothetical precursor molecule paleomycin has been predicted. To demonstrate the validity of the prediction, the paleomycin BGC has been synthesized and expressed in a bacterial strain providing the appropriate genetic background for turning the precursor peptide into a bioactive compound via various late-stage modifications. It is highly remarkable that production of several paleomycin variants was indeed detected by LC-MS/MS. Recombination and deletion events on the genetic level are described that have progressively converted paleomycin into the extant set of natural products. Interestingly, the evolution of A domains through point mutation could be demonstrated in combination with crystal structures, some of them containing amino acid ligands.

This highly interdisciplinary work is extremely interesting and expected to draw considerable attention. The reconstruction of an ancient natural product is a formidable achievement that is backed up with an impressive amount of data. Publication in *Nat Commun* is strongly recommended, when some weaknesses in the presentation and clarity of the manuscript have been taken care of.

- L42: Most relevant antibiotics are natural products, but let's not forget fluoroquinolones and sulfonamides. "Almost exclusively" should be toned down. Apart from aloe vera and the like, this reviewer is not aware of a single useful antibiotic derived from plants.
- Figure 1: Chemical structures must show consistent stereochemistry. In actinoidin and vancomycin, the amino acid on the right has no stereochemistry. In kistamycin, there is a bond missing. The sugars in vancomycin at the bottom right have no stereochemistry. The use of colors could be improved by not using the same color for different objects (e.g. Type IV GPA and Dpg have the same color). Fonts should have a readable size.
- Figure 3: Would it be possible to maybe use a more schematic representation for the complicated chemical structures everywhere except in Figure 1 (left)? The peptides on the assembly line in Figure 1 (right) are nicely presented. Perhaps, a strongly improved version of Figure 3 could be made where chemical changes are clearly correlated with genetic changes when such schematic drawings are used.
- Please decide whether to call the different clades by "Type I, II ..." or by the compound names ("Van-type"). This is inconsistent throughout the manuscript and makes it hard to follow. Perhaps, compound abbreviations would be preferable.
- Figure 4: The sum formulas are too small. The chemical structure is distorted and has a different format than the other chemical structures. "R3: =O" in the table does not work if R3 is defined in the structure as having a single bond to carbon.
- L161: The sequence of the reconstructed paleomycin BGC should be provided.
- L276 – L335: This section is extremely hard to follow because the A domains are apparently named inconsistently.
- For Figure 6, it might improve clarity if backbone atoms (C, N, O) were left out. Since the work of Conti et al., A domain binding pockets have usually been shown with the amino/carboxylate groups sticking out on top, which intuitively makes sense. Please specify clearly what is shown in panels E-G.
- The convention for kinetic parameters such as k_{cat} is to typeset the "k" in italics, not the subscript.
- Please clarify the sentence in L321: "These two key substitutions ..."
- Stereoindicators D and L are by convention set in small caps (throughout the text).
- L373 – 376: The logic in this sentence is not clear.
- L388: Why do the authors propose that the Shikimate products are metabolically more costly to produce? What evidence is this statement based on?
- SI P9: "to a r a" ?
- Table S1: Entries starting with "uncultured" should be checked. The strain names are confusing.

Reviewer #2 (Remarks to the Author):

In this very interdisciplinary and elegant study the groups of, Stegmann, Cryle, and Ziemert reconstitute an ancestral natural product of the glycopeptide family. A bioinformatics-based approach allowed to retrace the evolutionary events in order to infer the ancestral nonribosomal peptide synthetase that would produce a parental compound termed paleomycin. These findings are validated in a second step using a synthetic biology approach based on gene synthesis and heterologous expression. In-depth mass spectrometry analysis and bioactivity studies are used to gain more information about the structure and function of paleomycin. Additionally, the authors investigated the evolution of the substrate specificity of corresponding GPA NRPS A domains and experimentally validated their findings by A domain specificity assays and X-Ray crystallography. This study is a very nice combination of bioinformatics-based tools and wet-lab validations. The reader learns about evolutionary aspects of natural product evolution and the authors importantly validate these findings by heterologous expression, characterization of the ancestral natural product, and in-depth biochemical and crystallographic A domain analyses. This study should be of great interest to the readers of Nature Communications.

The in silico and 'wet lab' work is well performed, however, the documentation of the bioinformatics part should be more detailed.

Here is a summary of points that should be considered:

- Please use the same formatting and size of the molecular structures in the figures of the MS. In some cases, the resolution was poor, and the structures were difficult to see. In Figure 4 the carbohydrate moieties (R1) should be redrawn so that substituents are in axial and equatorial positions. Please provide figures in higher resolution.
- Please provide the code for all bioinformatics analyses that were done to create the ancestral sequences and presented figures in a common public repository, e.g. on github/zenodo/figshare, and add a link to it to the Data availability section
- I would suggest using the term ancestral or parental natural product instead of ancient natural product, e.g. Line 37: replace 'revive these ancient molecules' by 'revive the predicted ancestral molecules', Line 140: replace 'ancient GPAs' by 'ancestral GPAs', Line 351: replace 'revive an ancient antibiotic' by 'revive the potential ancestral antibiotic', and Line 405 replace "ancient" by "ancestral". The term ancient is often used in the context of ancient DNA (aDNA). The strength of this study as presented really lies in the identification of a phylogenetically ancestral molecule and not a prehistoric or ancient molecule that would be linked to a certain period in time. The authors should, however, cite the recent work to access ancient natural products using the complementary approach based on aDNA (regarding citations a few citations on A domain specificity would also help the reader find some relevant literature on that topic).
- Please use small capital letters for L/D specification of amino acids (e.g. line 283-289)
- Line 41: delete '(molecular weight < 2 kDa)'
- Line 57: change 'economically most' to 'most economically'
- Line 80: Unbold 'paleomycin'. In the same lines of the previous comment, ancestramycin (or something similar) would be a great alternative to the current name.
- Line 126, Figure 2: Why are 3 of the sequences in the gene tree not connected to a corresponding branch in the species tree? Please provide the corresponding species delineation for those gene branches. Please also provide information on the outgroup of the gene tree (is it a GPA BGC? In the text you write about 29 GPA BGCs but the tree shows 30 branch leaves) and bootstrap values on both trees. Multiple methods should be used to proof your conclusions. A 16S rRNA based species tree is not sufficient to make far-reaching statement about HGT, gene gain and loss. E.g. use MAFFT for Multiple core gene alignments or a whole genome-based phylogeny for creating a more precise species tree. Further, the result should be verified also on the amino acid level.
- Line 147&148: add 'likelihood for' e.g. '99.3% likelihood for...', '97.5% likelihood for ...'
- Line 167 & 180: Why did you pick a type III GPA strain with offline Bht biosynthesis and missing halogenase that had to be heavily modified instead of solely replacing the NRPS gene in a Teicoplanin producer or in *N. gerenzanensis*? Did you try to integrate the ancestral NRPS gene in a Teicoplanin producer or *N. gerenzanensis*?
- Line 182: If rpsD was deleted with the rpsA-D RistoKO, how can it be replaced again? Please make sure to correct typos in the gene abbreviations. Figure 4A shows that oxyD, rpsE and bhp

were replaced.

- Line 200: Do you have a structural suggestion for other important nodes, specifically for m/z 1177.26, 1339.31, and 1482.4?
- Line 201: Please indicate in Figure 4A the former location of the Ristomycin NRPS genes rpsA-D. Delete '(C)', put it as 'Figure 4. (A-C)' in Line 200.
- Line 204/205: Unbold 'HR-MS/MS based molecular network analysis', put (C) at the end of Line 207.

SI:

- SI Figure S38: What is the structure of F30 from *A. japonicum*? Is it similar/identical to one of the suggested paleomycin derivatives? Is F30 formed via the ancestral NRPS?
- Table S1: For reproducibility purposes please not only provide the genome accession numbers, but also the contig IDs and exact positions (start/end) of the GPA BGCs. Please also elaborate on how you identified the GPA BGC sequences (antiSMASH?).
- Table S3: Please also report the sequence of the ancestral paleomycin NRPS, possibly in a separate fasta file archived on zenodo/figshare. The chapter 'Sequence reconstruction' remains elusive in how exactly the ancestral paleomycin NRPS sequence was generated. Sharing of code and possible intermediate sequences would better allow us to understand how the sequence was bioinformatically predicted.

Reviewer #3 (Remarks to the Author):

Ziemert and colleagues use computational analyses of gene sequences encoding for NRPSs which encode GPAs to predict a hypothetical ancestral GPA-making NRPS and GPA, which they call paleomycin. They had the gene encoding this proposed paleomycin synthetase synthesized, and integrated it into metabolically engineered *A. japonicum*, to show the resulting engineered strain can make the paleomycin core. The authors then rationalized the putative evolutionary changes, and determined a set of structures of Acore constructs of one GPA synthetase (A1tei) with the specificity determining code amino acids of putative evolution intermediates mutated in (A1int, A1leu), as well as a D-Ala accepting Acore construct to document differences in substrate binding.

The study is interesting, though this reviewer does not have sufficient expertise in ancestral sequence reconstruction to critique that (major) component of the manuscript. The results do seem reasonable, but is it possible to state the paleomycin and paleomycin synthetase as reconstructed definitively existed exactly as predicted in this manuscript? As a non-expert, I cannot see how they can be definitively known, and believe the terms "putative" and "proposed" should be used liberally throughout this manuscript as descriptors of paleomycin and paleomycin synthetase.

In the statement:

The bioinformatically inferred DNA sequences of the ancestral nrps genes (27.859 kb, nrpsanc; with identities between 76% and 85% to the nrps genes of ristomycin (Table S4)) were synthesised in their entirety by ATG: synthetics (Merzhausen, Germany) and cloned into the p3SV vector, allowing integration into the surrogate host chromosome.

Do the percentages identity refer to nucleotide or amino acid sequence? Table S4 implies amino acid sequence (listing proteins, their % coverage and their % identity, but Table S5 and this statement implies nucleotide sequence. This is vital to clarify. Please state clearly the number of amino acid changes between the relevant synthetases, and how many amino acid changes are within ~8Å of active sites.

Figure 1 confuses me. Is the left panel and the right panel directly related? Is the color coding of

the left panel related to the color coding of the left panel? Is the left panel introduction and the right panel results (or indeed conclusion)? This needs to be re-worked (suggest splitting into two) and the figure legend made more accessible.

Were A1int and A1leu shown to be biochemically active? (Either as Acore or A domain?)

The metabolically engineering of *A. japonicum* DI_nrpsanc is quite involved. The authors:

-removed the ristomycin nrps genes rpsA-D

-added a second copy of the native StrR family regulator gene ajrR14 at the Φ C31 att site.

-replaced the "offline" Van-type Bht forming cassette (including the three genes oxyD, rpsD and bhaA) with that of a Tei-type non-heme iron oxygenase (hydroxylase) from *Nonomurea gerenzanensis* ATCC 39727

-introduced of the halogenase genes from *N. gerenzanensis* ATCC 39727.15,16

-introduced the NRPS gene

Please include a supplemental figure where all the changes are summarized (or do so in Fig 4, which seems incomplete in this regard). Were any experiment performed with less extensively engineered strains? For example, perhaps a different (cryptic or recognized) *A. japonicum* haologenase could act, rather than the introduced *N. gerenzanensis* halogenase.

Crystallography:

Structures 8GJ4, 8GIC, 8GJP and 8GKM are all from the same crystal form but only two of the deposited files seem to have consistency in designation of chain and which asymmetric unit is represented in the .cif file. Please make these uniform to facilitate the reader's viewing of the structures. Also, this hints that the MR was performed de novo for three or four of these four structures. Was this the case? Please specific in the Methods. If 1AMU was used the search model each time and new Rfree set was generated de novo with each MR, that's inefficient but technically fine. If one 8GJ4, 8GIC, 8GJP or 8GKM was used as a search model for phasing the others and a new Rfree set was generated, that biases the Rfree set.

Was the wavelength actually sometimes 0.9437 Å and sometimes 0.9436 Å?

The Rmerge for 8GLC is odd – why is it lower in the outer bin than overall? Please check into this.

The average multiplicity is reported incorrectly in for all structures. Please update Table S6.

For 8GJP, did the authors attempt to model waters? If not, why not? The data appear of sufficient quality to do so.

Minor

When was paleomycin and paleomycin synthetase predicted to exist? If not between ~550 and 250 million years ago, consider changing the name.

Both "nonribosomal" and "non-ribosomal" are used in the abstract. Make uniform. (I suggest "nonribosomal".)

Why are natural products defined as less than 2 kDa?

"usually include genes for precursor and core biosynthesis, regulation, resistance, and transport": Include post-core biosynthesis modification in this list.

We would like to thank all the reviewers and the editorial staff for their time in the consideration of our manuscript. We have answered or addressed all the points raised below in a manner that we hope will be to the reviewer's satisfaction. Our comments are written in red.

REVIEWER COMMENTS

Reviewer #1 (Remarks to the Author):

Hansen et al. have investigated the evolution of glycopeptide antibiotics which are important drugs of last resort. From various related gene clusters, a phylogeny has been built and an ancestral gene cluster for the biosynthesis of the hypothetical precursor molecule paleomycin has been predicted. To demonstrate the validity of the prediction, the paleomycin BGC has been synthesized and expressed in a bacterial strain providing the appropriate genetic background for turning the precursor peptide into a bioactive compound via various late-stage modifications. It is highly remarkable that production of several paleomycin variants was indeed detected by LC-MS/MS. Recombination and deletion events on the genetic level are described that have progressively converted paleomycin into the extant set of natural products. Interestingly, the evolution of A domains through point mutation could be demonstrated in combination with crystal structures, some of them containing amino acid ligands.

This highly interdisciplinary work is extremely interesting and expected to draw considerable attention. The reconstruction of an ancient natural product is a formidable achievement that is backed up with an impressive amount of data. Publication in Nat Commun is strongly recommended, when some weaknesses in the presentation and clarity of the manuscript have been taken care of.

Thank you for your comments and your assessment of our manuscript!

- L42: Most relevant antibiotics are natural products, but let's not forget fluoroquinolones and sulfonamides. "Almost exclusively" should be toned down. Apart from aloe vera and the like, this reviewer is not aware of a single useful antibiotic derived from plants.

We appreciate this comment, and have altered the sentence in the manuscript to the following: Natural products (~~molecular weight < 2 kDa~~) form one of the most important sources of medicinal compounds, with modern medicine reliant on antibiotics that ~~almost exclusively~~ that often originate from biosynthesis in various microorganisms ~~and plants~~.

- Figure 1: Chemical structures must show consistent stereochemistry. In actinoidin and vancomycin, the amino acid on the right has no stereochemistry. In kistamycin, there is a bond missing. The sugars in vancomycin at the bottom right have no stereochemistry. The use of colors could be improved by not using the same color for different objects (e.g. Type IV GPA and Dpg have the same color). Fonts should have a readable size.

Figures have been corrected, standardised and the stereochemistry is now shown for all – thank you for spotting this!

- Figure 3: Would it be possible to maybe use a more schematic representation for the complicated chemical structures everywhere except in Figure 1 (left)? The peptides on the assembly line in Figure 1 (right) are nicely presented. Perhaps, a strongly improved version of

Figure 3 could be made where chemical changes are clearly correlated with genetic changes when such schematic drawings are used.

We appreciate these comments, which have helped us to improve these important figures. We have introduced a schematic for the GPA structural figure (1) biosynthesis figure (2), and then include these into the revised figure 3 (now 4), where these are used to show the predicted structures of the products at key nodes noted in this figure.

- Please decide whether to call the different clades by “Type I, II ...” or by the compound names (“Van-type”). This is inconsistent throughout the manuscript and makes it hard to follow. Perhaps, compound abbreviations would be preferable.

We have standardised the manuscripts to using compound abbreviation nomenclature as suggested, and only retain the “type V” nomenclature as these constitute a significant outgroup not under investigation in this work.

- Figure 4: The sum formulas are too small. The chemical structure is distorted and has a different format than the other chemical structures. “R3: =O” in the table does not work if R3 is defined in the structure as having a single bond to carbon.

We have hopefully addressed this by remaking figure 4 (now 5) with these helpful suggestions in mind. The genetic engineering undertaken is now described in stepwise form in SI Figure S31 as suggested by reviewer 3.

- L161: The sequence of the reconstructed paleomycin BGC should be provided.

The sequence of the reconstructed paleomycin is now available as additional material on zenodo (DOI 10.5281/zenodo.8410710).

- L276 – L335: This section is extremely hard to follow because the A domains are apparently named inconsistently.

We apologise for the naming confusion in this section. We have attempted to simplify the naming such that all ancestral A1 domain proteins are labelled ANC1-4 (with no amino acid specificity indicated). When discussing the A1 domain from teicoplanin biosynthesis, we use the term A1_{tei}. For pocket grafts (where we include the mutations predicted from ANC2-3 in the substrate specificity pocket of A1_{tei}), we use the terms A1_{ANC2} and A1_{ANC3}. The residues within the substrate selection pockets of A1_{tei} and ANC1 are identical.

- For Figure 6, it might improve clarity if backbone atoms (C, N, O) were left out. Since the work of Conti et al., A domain binding pockets have usually been shown with the amino/carboxylate groups sticking out on top, which intuitively makes sense. Please specify clearly what is shown in panels E-G.

We have provided two alternative versions of this figure below, in which the view is rotated to that closer to what is seen in the 1AMU structure paper, and the other in which the mainchain atoms are hidden (except for Gly) and are happy to select whichever the reviewers feel helps with clarity of this figure. We have also attempted to more clearly specify the comparison of pockets that are contained in panels E-G.

- The convention for kinetic parameters such as k_{cat} is to typeset the “k” in italics, not the subscript.

Corrected.

- Please clarify the sentence in L321: “These two key substitutions ...”

We have added (H237Y and L295V) to clarify the residues we are discussing here.

- Stereoindicators D and L are by convention set in small caps (throughout the text).

Corrected.

- L373 – 376: The logic in this sentence is not clear.

We have altered this section to the following to improve clarity here:

The replacement of the ancestral GPA M3 by an Asn-encoding module via recombination was observed simultaneously with the fusion event involving M2 and M3. This observation is consistent with experimental evidence that the fusion of M2 and M3 within natural GPA assembly lines results in the subsequent loss of peptide extension.

- L388: Why do the authors propose that the Shikimate products are metabolically more costly to produce? What evidence is this statement based on?

Our logic is based on ability of a bacterium to produce more Van-type GPA than a Tei-type GPA from the same pool of Shikimate precursors.

- SI P9: “to a r a” ?

Thank you for spotting this typo – we have corrected it.

- Table S1: Entries starting with “uncultured” should be checked. The strain names are confusing.

Thank you for spotting this. Here two strains and accession numbers were accidentally placed in a row. We have corrected this.

Reviewer #2 (Remarks to the Author):

In this very interdisciplinary and elegant study the groups of, Stegmann, Cryle, and Ziemert reconstitute an ancestral natural product of the glycopeptide family. A bioinformatics-based approach allowed to retrace the evolutionary events in order to infer the ancestral nonribosomal peptide synthetase that would produce a parental compound termed paleomycin. These findings are validated in a second step using a synthetic biology approach based on gene synthesis and heterologous expression. In-depth mass spectrometry analysis and bioactivity studies are used to gain more information about the structure and function of paleomycin. Additionally, the authors investigated the evolution of the substrate specificity of corresponding GPA NRPS A domains and experimentally validated their findings by A domain specificity assays and X-Ray crystallography.

This study is a very nice combination of bioinformatics-based tools and wet-lab validations. The reader learns about evolutionary aspects of natural product evolution and the authors importantly validate these findings by heterologous expression, characterization of the ancestral natural product, and in-depth biochemical and crystallographic A domain analyses. This study should be of great interest to the readers of Nature Communications.

Thank you very much for your review of our manuscript!

The in silico and ‘wet lab’ work is well performed, however, the documentation of the bioinformatics part should be more detailed.

Here is a summary of points that should be considered:

- Please use the same formatting and size of the molecular structures in the figures of the MS. In some cases, the resolution was poor, and the structures were difficult to see. In Figure 4 the carbohydrate moieties (R1) should be redrawn so that substituents are in axial and equatorial positions. Please provide figures in higher resolution.

We apologise for the low resolution of the figures in the original submission. These were 600 dpi PNGs but there was clearly an issue with pdf generation. We have substantially altered the figures in line with all of the reviewer’s comments, including standardising all chemical structures, and addressed the comments specifically mentioned here.

- Please provide the code for all bioinformatics analyses that were done to create the ancestral sequences and presented figures in a common public repository, e.g. on github/zenodo/figshare, and add a link to it to the Data availability section

Supporting data (paleomycin sequence, sequence alignments, phylogenetic trees, bioinformatic scripts, posterior probabilities for the ancestral sequence reconstruction, sliding window analysis) have been deposited at zenodo (DOI 10.5281/zenodo.8410710).

- I would suggest using the term ancestral or parental natural product instead of ancient natural product, e.g. Line 37: replace ‘revive these ancient molecules’ by ‘revive the predicted ancestral molecules’, Line 140: replace ‘ancient GPAs’ by ‘ancestral GPAs’, Line 351: replace ‘revive an ancient antibiotic’ by ‘revive the potential ancestral antibiotic’, and Line 405 replace “ancient” by “ancestral”. The term ancient is often used in the context of ancient DNA (aDNA). The strength of this study as presented really lies in the identification of a phylogenetically ancestral molecule and not a prehistoric or ancient molecule that would be linked to a certain period in time. The authors should, however, cite the recent work to access ancient natural products using the complementary approach based on aDNA (regarding citations a few citations on A domain specificity would also help the reader find some relevant literature on that topic).

We have altered the use of the word ancient to ancestral as suggested, which was an excellent suggestion. We have also introduced the following citations to discuss aDNA (Klapper et al, Science, 2023) and A-domain specificity as suggested (Stachelhaus et al, Chemistry & Biology, 1999; Stanišić & Kries, ChemBioChem, 2019).

- Please use small capital letters for L/D specification of amino acids (e.g. line 283-289)

These have been altered to small caps.

- Line 41: delete ‘(molecular weight < 2 kDa)’

Deleted.

- Line 57: change ‘economically most’ to ‘most economically’

Fixed.

- Line 80: Unbold 'paleomycin'. In the same lines of the previous comment, ancestramycin (or something similar) would be a great alternative to the current name.

The text for paleomycin has altered to italics only. We appreciate the comment regarding naming, however, also feel that the terminology we've selected for the putative ancestral GPA is appropriate and would prefer to keep this naming.

- Line 126, Figure 2: Why are 3 of the sequences in the gene tree not connected to a corresponding branch in the species tree? Please provide the corresponding species delineation for those gene branches. Please also provide information on the outgroup of the gene tree (is it a GPA BGC? In the text you write about 29 GPA BGCs but the tree shows 30 branch leaves) and bootstrap values on both trees. Multiple methods should be used to proof your conclusions. A 16S rRNA based species tree is not sufficient to make far-reaching statement about HGT, gene gain and loss. E.g. use MAFFT for Multiple core gene alignments or a whole genome-based phylogeny for creating a more precise species tree. Further, the result should be verified also on the amino acid level.

Figure 2 (now 3) was completely reworked in line with the reviewers' suggestions. We now use a multi locus sequence tree (alignment of multiple core genes) as the species tree. We further added a section to the material and methods section to describe the tree building methods. Here we also state why two branches of the gene tree are not connected to the species tree. The full tree, including the corresponding species delineation is provided as supplementary data on zenodo (DOI 10.5281/zenodo.8410710).

- Line 147&148: add ,likelihood for' e.g. '99.3% likelihood for...', '97.5% likelihood for ...'

Added.

- Line 167 & 180: Why did you pick a type III GPA strain with offline Bht biosynthesis and missing halogenase that had to be heavily modified instead of solely replacing the NRPS gene in a Teicoplanin producer or in *N. gerenzanensis*? Did you try to integrate the ancestral NRPS gene in a Teicoplanin producer or *N. gerenzanensis*?

We selected *A. japonicum* as our preferred choice due to its genetic manipulability, rapid growth, dispersal, and sporulation capabilities, as well as its possession of the *vanHAX* resistant genes - a feature absent in other GPA producers, such as *Nonomuraea*. In previous experiments, we have indeed worked with the other strains. However, the cultivation and the genetic manipulation proved considerably more challenging in these cases. Consequently, we made the strategic decision to optimize *A. japonicum* as our heterologous host and did not use the other producer strains.

- Line 182: If *rpsD* was deleted with the *rpsA-D* RistoKO, how can it be replaced again? Please make sure to correct typos in the gene abbreviations. Figure 4A shows that *oxyD*, *rpsE* and *bhp* were replaced.

Thank you for spotting this. We exchanged in line 182 "*oxyD*, *rpsD* and *bhaA*" in "*oxyD*, *rpsE* and *bhp*"

- Line 200: Do you have a structural suggestion for other important nodes, specifically for m/z 1177.26, 1339.31, and 1482.4?

Based upon the biosynthesis of related GPAs (such as teicoplanin), these nodes (with -16 Da) are indicative of a Tyr residue in place of a Bht residue. Whilst this could be at either position 2 or 6 of the peptide, comparison with the products of these more closely related clusters would support this being Tyr-2. However, we are unable to unequivocally assign this due to the limited fragmentation seen with such heavily crosslinked peptides.

- Line 201: Please indicate in Figure 4A the former location of the Ristomycin NRPS genes rpsA-D. Delete '(C)', put it as 'Figure 4. (A-C)' in Line 200.

Based on the comments of Reviewer 3 we have split the original Figure 4, with the genetic manipulation now included in a far more detailed SI figure (SI Figure S31) and the remaining figure corrected as suggested by the reviewers. We have therefore also altered the call outs to this figure (now figure 5) in the main text.

- Line 204/205: Unbold 'HR-MS/MS based molecular network analysis', put (C) at the end of Line 207.

The font has been corrected, and the panel names are no longer used for this figure.

SI:

- SI Figure S38: What is the structure of F30 from *A. japonicum*? Is it similar/identical to one of the suggested paleomycin derivatives? Is F30 formed via the ancestral NRPS?

No peaks could be detected in fraction 30 in *A. japonicum* Δ nrps. Only when the ancestral NRPS were introduced into the strain were peaks identified in fraction 30.

- Table S1: For reproducibility purposes please not only provide the genome accession numbers, but also the contig IDs and exact positions (start/end) of the GPA BGCs. Please also elaborate on how you identified the GPA BGC sequences (antiSMASH?).

The BGCs for the GPAs were identified with antiSMASH, which is now indicated in the Materials and Methods section of the SI.

- Table S3: Please also report the sequence of the ancestral paleomycin NRPS, possibly in a separate fasta file archived on zenodo/figshare. The chapter 'Sequence reconstruction' remains elusive in how exactly the ancestral paleomycin NRPS sequence was generated. Sharing of code and possible intermediate sequences would better allow us to understand how the sequence was bioinformatically predicted.

The sequence for the ancestral paleomycin NRPS is also now uploaded and available on zenodo (DOI 10.5281/zenodo.8410710).

Reviewer #3 (Remarks to the Author):

Ziemert and colleagues use computational analyses of gene sequences encoding for NRPSs

which encode GPAs to predict a hypothetical ancestral GPA-making NRPS and GPA, which they call paleomycin. They had the gene encoding this proposed paleomycin synthetase synthesized, and integrated it into metabolic engineered *A. japonicum*, to show the resulting engineered strain can make the paleomycin core. The authors then rationalized the putative evolutionary changes, and determined a set of structures of Acore constructs of one GPA synthetase (A1tei) with the specificity determining code amino acids of putative evolution intermediates mutated in (A1int , A1leu), as well as a D-Ala accepting Acore construct to document differences in substrate binding.

The study is interesting, though this reviewer does not have sufficient expertise in ancestral sequence reconstruction to critique that (major) component of the manuscript. The results do seem reasonable, but is it possible to state the paleomycin and paleomycin synthetase as reconstructed definitively existed exactly as predicted in this manuscript? As I non-expert, I cannot see how they can be definitively known, and believe the terms “putative” and “proposed” should be used liberally throughout this manuscript as descriptors of paleomycin and paleomycin synthetase.

Thank you for your review of our manuscript and for this comment – we agree and have now included these labels frequently throughout the revised version.

In the statement:

The bioinformatically inferred DNA sequences of the ancestral nrps genes (27.859 kb, nrpsanc; with identities between 76% and 85% to the nrps genes of ristomycin (Table S4)) were synthesised in their entirety by ATG:synthetics (Merzhausen, Germany) and cloned into the p3SV vector, allowing integration into the surrogate host chromosome. Do the percentages identity refer to nucleotide or amino acid sequence? Table S4 implies amino acid sequence (listing proteins, their % coverage and their % identity, but Table S5 and this statement implies nucleotide sequence. This is vital to clarify.

To remove potential confusion, we have deleted Table S5 as it is currently included.

Please state clearly the number of amino acid changes between the relevant synthetases, and how many amino acid changes are within ~8Å of active sites.

The amino acid changes between the paleomycin biosynthesis genes and the ristomycin biosynthesis genes are stated in Table S4. We have included the amino acid selection (Stachelhaus) codes for the main GPA types as a replacement SI Table S5.

Figure 1 confuses me. Is the left panel and the right panel directly related? Is the color coding of the left panel related to the color coding of the right panel? Is the left panel introduction and the right panel results (or indeed conclusion)? This needs to be re-worked (suggest splitting into two) and the figure legend made more accessible.

We have taken this suggestion on board (also in keeping with the comments from reviewer 1) and have split this figure now into two, one including the different structures and their schematic representations (Figure 1) and the biosynthetic assembly line (Figure 2).

Were A1int and A1leu shown to be biochemically active? (Either as Acore or A domain?)

We only generated Acore constructs for these pocket graft constructs to assess the structural impacts of the key selection pocket mutations identified from our bioinformatic analyses; the activity presented in figure 6 (now 7) is from the full A-domains at the predicted nodes shown in 7A along with the A1 domain from teicoplanin biosynthesis.

The metabolically engineering of *A. japonicum* DI_nrpsanc is quite involved. The authors:
-removed the ristomycin nrps genes rpsA-D
-added a second copy of the native StrR family regulator gene ajrR14 at the Φ C31 att site.
-replaced the “offline” Van-type Bht forming cassette (including the three genes oxyD, rpsD and bhaA) with that of a Tei-type non-heme iron oxygenase (hydroxylase) from *Nonomurea gerenzanensis* ATCC 39727
-introduced of the halogenase genes from *N. gerenzanensis* ATCC 39727.15,16
-introduced the NRPS gene
Please include a supplemental figure where all the changes are summarized (or do so in Fig 4, which seems incomplete in this regard).

We have taken these comments onboard and have removed the highly truncated scheme from figure 4 (now 5). This has now been incorporated into a new SI figure (SI Figure S31) that we hope shows the workflow far more clearly.

Were any experiment performed with less extensively engineered strains? For example, perhaps a different (cryptic or recognized) *A. japonicum* haologenase could act, rather than the introduced *N. gerenzanensis* halogenase.

When we initially introduced the genes into *A. japonicum* Dnrps, no production was observed. We conducted production tests after each individual modification was introduced, but it was only when all modifications were implemented that we observed production. It is worth noting that the halogenases responsible for GPA production exhibit a high degree of specificity. Even if *A. japonicum* possesses other halogenases, they are unable to halogenate GPA. Additionally, we have never observed the presence of halogenated ristomycin.

Crystallography:

Structures 8GJ4, 8GIC, 8GJP and 8GKM are all from the same crystal form but only two of the deposited files seem to have consistency in designation of chain and which asymmetric unit is represented in the .cif file. Please make these uniform to facilitate the reader’s viewing of the structures. Also, this hints that the MR was performed de novo for three or four of these four structures. Was this the case? Please specific in the Methods. If 1AMU was used the search model each time and new Rfree set was generated de novo with each MR, that’s inefficient but technically fine. If one 8GJ4, 8GIC, 8GJP or 8GKM was used as a search model for phasing the others and a new Rfree set was generated, that biases the Rfree set.

Regarding the concerns about molecular replacement (MR). For the initial structure, 8GJ4, we used 1AMU as the search model. Subsequently, Chain A and D from the final model of 8GJ4 served as search models for 8GIC, 8GJP, and 8GKM.

To avoid potential R-free bias, we used PDB tools to randomize ADPs, followed by simulated annealing to ensure the effective removal of any “memory” of the original R-free flags. We have now added extra detail in the Methods section for clarity.

Was the wavelength actually sometimes 0.9437 Å and sometimes 0.9436 Å?

These are the values reported from the beamline; presumably the wavelength was very close to 0.94365 Å and thus minor changes lead to the difference in rounding.

The Rmerge for 8GLC is odd – why is it lower in the outer bin than overall? Please check into this.

Thank you for pointing out the inconsistency in the Rmerge values for 8GLC. Upon revisiting our data processing steps, we identified the oversight. This arose from an error in supplying the wrong input file to PHENIX during Table 1 generation. We've corrected this, updated Table 1. We appreciate your vigilance and apologise for any confusion this may have caused.

The average multiplicity is reported incorrectly in for all structures. Please update Table S6.

Thank you for noting the discrepancy in average multiplicity. This error arose from the same oversight previously mentioned, where an incorrect input file was used in PHENIX. We have now corrected and updated Table S6. We apologize for the oversight and appreciate your diligence.

For 8GJP, did the authors attempt to model waters? If not, why not? The data appear of sufficient quality to do so.

The density present in 8JGP did not support the modelling of waters for this structure.

Minor

When was paleomycin and paleomycin synthetase predicted to exist? If not between ~550 and 250 million years ago, consider changing the name.

The putative paleomycin synthetase was predicted to exist in the Paleozoic era (550 – 250 mya). According to Wright et al (#ref 12) the ancestral GPA scaffold evolved 500 - 300 mya, which agrees with our analysis. Furthermore, Wright and co-workers dated the *Amycolatopsis* clade back to 303 mya: since our ancestral GPA has evolved earlier than the *Amycolatopsis* GPA scaffold we should therefore be within this range.

Both “nonribosomal” and “non-ribosomal” are used in the abstract. Make uniform. (I suggest “nonribosomal”.)

We have standardised the nomenclature to nonribosomal as suggested.

Why are natural products defined as less than 2 kDa?

This has been removed, also at the suggestion of the other reviewers.

“usually include genes for precursor and core biosynthesis, regulation, resistance, and transport”: Include post-core biosynthesis modification in this list.

We have added this as suggested.